# The psychophysics of bouncing: Perceptual constraints, physical constraints, animacy, and phenomenal causality

**Michele Vicovaro**[1], **Loris Brunello**[1], **Giulia Parovel**[2]*

**1** Department of General Psychology, University of Padova, Padova, Italy, **2** Department of Social, Political and Cognitive Sciences, University of Siena, Siena, Italy

* giulia.parovel@unisi.it

**Data Availability Statement:** Data and research material are available on OSF at the following link: osf.io/xz8t3.

## Abstract

In the present study we broadly explored the perception of physical and animated motion in bouncing-like scenarios through four experiments. In the first experiment, participants were asked to categorize bouncing-like displays as physical bounce, animated motion, or *other*. Several parameters of the animations were manipulated, that is, the simulated coefficient of restitution, the value of simulated gravitational acceleration, the motion pattern (uniform acceleration/deceleration or constant speed) and the number of bouncing cycles. In the second experiment, a variable delay at the moment of the collision between the bouncing object and the bouncing surface was introduced. Main results show that, although observers appear to have realistic representations of physical constraints like energy conservation and gravitational acceleration/deceleration, the amount of visual information available in the scene has a strong modulation effect on the extent to which they rely on these representations. A coefficient of restitution >1 was a crucial cue to animacy in displays showing three bouncing cycles, but not in displays showing one bouncing cycle. Additionally, bouncing impressions appear to be driven by perceptual constraints that are unrelated to the physical realism of the scene, like preference for simulated gravitational attraction smaller than *g* and perceived temporal contiguity between the different phases of bouncing. In the third experiment, the visible opaque bouncing surface was removed from the scene, and the results showed that this did not have any substantial effect on the resulting impressions of physical bounce or animated motion, suggesting that the visual system can fill-in the scene with the missing element. The fourth experiment explored visual impressions of causality in bouncing scenarios. At odds with claims of current causal perception theories, results indicate that a passive object can be perceived as the direct cause of the motion behavior of an active object.

## Introduction

*Then he squatted down, held the ball about a half-inch from the floor, dropped it.*

**Funding:** This publication was made possible thanks to the specific contribution of the University of Siena (Italy) for Open Access support and the PSR funding of the Department of Social, Political and Cognitive Sciences (DISPOC), University of Siena (Italy). The funders had no role in study design, data collection and analysis, decision to publish, or preparation of the manuscript.

**Competing interests:** The authors have declared that no competing interests exist.

*It bounced, naturally enough. Then it bounced again. And again. Only this was not natural, for on*

*the second bounce the ball went higher in the air than on the first, and on the third bounce higher*

*still. After a half minute, my eyes were bugging out and the little ball was bouncing four feet in the*

*air and going higher each time.*

*I grabbed my glass. "What the hell!" I said.*

(W. Tevis, *The Big Bounce*, 1958)

If we are asked to represent the typical behavior of a bouncing ball, we would probably imagine a ball falling down, bouncing against a surface, and then moving back off repeatedly with a decreasing speed and a lower peak height after each bounce, until stillness. Instead, we would be really surprised, or even amused, if we saw a ball increasing its speed and peak height after each bounce, as if it were jumping (see the incipit, and see this video at: https://youtu.be/oQs5WJLUFEM). Apparently, we do not need knowledge of energy conservation principles to perceive a bounce or a jump, like we do not need knowledge of the physics of light to perceive colors.

From a perceptual standpoint, bouncing can be defined as an *event*, that is, as a sequence of motions with a definite, meaningful perceptual structure. A prominent feature of visually perceived events is phenomenal causality, that is, the impression that the motions of the objects in the scene are causally related [1]. For instance, in Michotte's [2] *launching effect*, observers are presented with two horizontally aligned squares, and at a point in time one square (*A*) starts moving towards the other (*B*). When *A* touches *B*, *B* starts moving with the same velocity as *A*, whilst *A* came to a stop. This simple configuration leads to the vivid visual impression that *A* launches or kicks *B*, that is, that the motion of *B* is caused by the collision with *A*. The phenomenon is impervious to observers' explicit knowledge and expectations, which provides support to its perceptual (rather than post-perceptual) origins [3–6]. As another example of visual event, White and Milne [7] presented the participants with simple 2D animations showing an object (*A*) moving towards a set of initially stationary elements that, upon contact with *A*, started moving in different directions. Under appropriate conditions, this stimulus configuration led to the vivid impressions of *enforced disintegration* or *bursting*. Other examples of visually perceived events include triggering, entraining, expulsion [2], braking [8], penetration [9], pulling [10], generative transmission [11], intentional reaction [12], and shattering [13].

According to Gestalt-theoretic accounts, visual perception of events depends on specific perceptual principles that are unrelated to physical laws, such as motion ampliation [2], coincidence avoidance [14, 15], and grouping [16, 17]. Within the Gestalt-theoretic perspective, various authors have upheld the hypothesis that the laws underlying the visual perception of events are independent of physical laws and of the observer's prior experience with the physical environment (see also [2, 6, 18, 19]). In this perspective, physical realism would be neither necessary nor sufficient for the visual perception of events.

Other authors have instead emphasized the possible role of past experience in shaping the visual perception of events and the related visual impressions of causality. For instance, according to White's [20, 21] schema-matching model, information in the visual stimulus would be matched to schemas acquired through everyday perceptual-motor experience with physical objects. Subjective motor experiences of forces do not faithfully represent the

corresponding physical forces, which explains why the laws underlying visually perceived events may significantly depart from the predictions of Newtonian physics (e.g., at odds with Newton's third law, forces generated by collisions are perceived to be asymmetrical [20, 22]). In a similar vein, Hubbard [23] has suggested that causality would be perceived when the visual stimulus matches an impetus transmission heuristic, by which an agent (e.g., *A* in the launching effect) is seen to impart or transmit an impetus to a patient (e.g., *B*; see also [24]). The impetus transmission heuristic would be acquired from motor experience with physical forces (see also [25]). Two noteworthy characteristics of the impetus transmission heuristic are that the impetus is conserved (i.e., the impetus of the patient cannot be greater than the impetus transmitted to it by the agent) and that impetus dissipates, leading to a gradual slowdown of the patient.

Models based on schema-matching and on impetus transmission heuristic emphasize the discrepancies between the laws underlying the visual perception of events and the corresponding Newtonian laws. By contrast, according to the *noisy Newton* model, the perception and the interpretation of physical events would be based on internalized physical laws [26–30]. According to this model, events perception can be reconciled with the predictions from Newtonian mechanics, if uncertainty due to sensory noise is taken into account. However, Kominsky et al. [31] argued that perceptual constraints may impose limits on the number of physical constraints that can actually be internalized, and on the precision with which they are represented. For instance, in launching-like scenarios, representations of the range of physically possible speed ratios between *A* and *B* would be imprecise due to the limited sensitivity of the visual system to speed variations.

The first aim of the present work was to explore the laws underlying the visual perception of physical motion and animated motion in bouncing-like scenarios, and to provide a systematic comparison between these laws and the corresponding physical laws. Specifically, Experiments 1 and 2 explored how the visual perception of physical motion and animated motion are affected by a variety of physical parameters (i.e., coefficient of restitution, motion pattern, simulated value of gravitational acceleration, number of bouncing cycles, presence of a time delay at the moment of the impact between the bouncing object and the bouncing surface). The results can provide novel insights about the possible interplay between physical and perceptual constraints in shaping the visual perception of events, and more in general they can inform the debate around current theories of causal perception. In Experiment 3, we tested the necessity of an opaque visible bouncing surface for the visual perception of physical motion and animated motion in bouncing-like scenarios. Lastly, Experiment 4 explored if and how visual impressions of physical motion and animated motion are related to visual impressions of causality.

## The physics of bouncing

We focus on the behavior of an object that falls vertically on a rigid horizontal surface [32]. For simplicity, hereafter we refer to a *bouncing cycle* as a sequence of events formed by 1) a falling phase, in which the object falls downwards under the effects of gravitational acceleration; 2) a collision against a rigid bouncing surface; 3) a climbing-back phase in which the objects move upwards (again, under the effects of gravitational acceleration) until stillness.

In the *falling phase*, the object accelerates downwards due to the effects of gravity. Ignoring the possible effects of air resistance, the object's acceleration is $g = 9.80665 \approx 9.81$ $m/s^2$, and its velocity just before the collision with the surface ($v_0$) is given by:

$$v_0 = \sqrt{2gh_0}, \tag{1}$$

where $h_0$ is the object's initial height from the surface (i.e., the falling height). The duration $t_0$ of the falling phase is

$$t_0 = \sqrt{\frac{2h_0}{g}}. \tag{2}$$

Upon contact with the surface, the object compresses and then expands, bouncing back off the surface and moving upwards. The object's *collision* behavior depends on a complex interaction between its material properties, the material properties of the surface, the object's shape and its possible spin [33]. Ignoring these complexities, the object's bouncing behavior is well described by Newton's Law of Restitution, according to which

$$v_1 = -(v_0 \times C), \tag{3}$$

where $v_1$ is the object's post-collision velocity and $C$ is the *coefficient of restitution*. Here the minus sign represents the inversion of the motion direction after the collision with the surface. In terms of energy conservation and dissipation, upon the collision with the surface, the initial kinetic energy of the falling object is converted to elastic potential energy, which is then converted back to kinetic energy. Because of energy conservation principles, the kinetic energy after the collision (i.e., $0.5mv_1^2$) cannot be greater than the kinetic energy before the collision (i.e., $0.5mv_0^2$), which means that $C$ cannot be greater than 1. In the energy transformation processes, some energy is always dissipated as heat, therefore $C$ is typically smaller than 1. Moreover, $C$ cannot be smaller than 0, otherwise the post-collision velocity would have the same direction as the pre-collision velocity (i.e., the object would penetrate inside the surface). All this implies $0 \leq C \leq 1$, that is, $0 \leq |v_1| \leq v_0$.

The coefficient of restitution $C$ constitutes a link between the material properties of the bouncing object and its motion pattern [34]. Relatively inelastic objects such as balls made of plasticine or clay are associated with small values of $C$, and tend to stick to the surface after the collision (i.e., high energy dissipation; low post-collision speed and peak height). On the contrary, relatively elastic objects such as tennis balls and superballs are associated with high values of $C$, and tend to bounce high after the collision with the surface (i.e., low energy dissipation; high post-collision speed and peak height).

In the *climbing-back phase*, the object decelerates until it reaches a peak height $h_1$ given by

$$h_1 = \frac{v_1^2}{2g^2}, \tag{4}$$

The duration $t_1$ of the climbing-back phase is

$$t_1 = \sqrt{\frac{2h_1}{g}}. \tag{5}$$

After the first bounce, a new bounce starts, and $h_1$ is the starting position of the object in the second falling phase. A variable number of bounces may occur, until the object remains still on the surface.

## The intuitive physics of bouncing

In the case of bouncing, the motion pattern of the bouncing object is uniquely related to parameter $C$, and therefore to the elasticity of the bouncing object itself. More specifically, $C$ is specified by the ratio of the velocities before and after the collision, by the ratio of the peak

heights, and by the ratio of the periods of successive bounces:

$$C = -(v_1/v_0) = \sqrt{h_1/h_0} = \tau_1/\tau_0, \tag{6}$$

where $\tau_0$ is the period of the first bounce (i.e., $t_0+t_1$) and $\tau_1$ is the period of the second bounce [35].

Previous studies on bouncing perception have mainly focused on observers' ability to perceive the elasticity of the bouncing object from the bounce kinematics. Warren et al. [35] showed that participants could accurately estimate the elasticity of a simulated bouncing object (i.e., a simulated disk falling vertically downwards and bouncing various time on a flat surface), mainly relying on the ratio of peak heights. Similar results have also been reported by Nusseck et al. [36] and Paulun and Fleming [37] in more realistic bouncing scenarios.

Elasticity ratings can be used to explore participants' intuitive knowledge of the relationship between the object's elasticity and its kinematics, but they do not provide information about whether observers can discriminate between physically possible and physically impossible bounces. However, in their study, Paulun and Fleming [37] also asked participants to rate the typicality of the motion of virtual bouncing cubes, and reported a negative relationship between elasticity and typicality. In other words, there would be a realistic perceptual prior for relatively inelastic objects in the real world.

Twardy and Bingham [38] presented participants with a simulated small-scale 3D environment showing a sphere falling to the ground from high above with a parabolic trajectory and bouncing against the simulated ground multiple times. Results showed that the naturalness ratings were relatively high for $C \leq 1$ (with a peak for $C = 0.9$), whereas, consistently with physics, motions characterized by $C > 1$ were judged as unnatural. In addition, participants provided low naturalness ratings to animations showing decreasing gravity, whereas they were relatively insensitive to gravity increases. These results indicate that conservation violations are perceptually unnatural.

Previous studies on the intuitive physics of bouncing have used relatively realistic virtual scenarios showing an object moving in a 3D environment [37, 38]. However, converging evidence indicates that events can be unambiguously perceived even in the case of schematic 2D animations showing simple shapes moving on a uniform background (for a review see [1]). As a demonstration of their being deeply rooted in early visual processing, studies have shown that visually perceived events can influence visual memory [24, 39] and low-level spatiotemporal properties of the scene, such as distance [40, 41], trajectory [42], speed [43, 44], and timing [45].

## From physical to animated motion

So far, we have focused on visual impressions involving the motion of inanimate objects. However, under appropriate conditions, animations may give rise to vivid impressions of *animacy*. For instance, when the stimuli that give rise to the launching effect are modified so that the post-collision velocity of *B* is more than two times as large as the pre-collision velocity of *A*, the launching effect leaves place to the so-called *triggering effect* [2, 18]. In this case, the observers have the impression that the motion of *B* is triggered by the contact with *A*, however the motion of *B* is perceived as active, that is self-generated by *B* and not passively caused by the collision with *A* as in the launching effect. Moreover, when *B* starts moving before the collision with *A* and faster than it, observers have the impression of an *intentional reaction* in which *B* is seen as self-propelling and intentionally escaping at the arrival of *A* [12]. In these cases, a *hidden inner force* is attributed to *B*, that makes it move independently of the force transmitted to it by *A*.

Since these pioneering works, many authors have argued that the perception of animacy is triggered by the manipulation of spatiotemporal contingencies between two or more moving objects, or between a moving object and other contextual elements (e.g., [46–50]). A general factor that triggers animacy impressions would be the apparent violations of Newtonian principles (e.g., [51, 52]). Other simpler cues to animacy are self-propulsion [53, 54] and non-rigid, rhythmic expansion-contraction motion [2, 55, 56]. According to several authors, phenomena like causality and animacy, despite the seemingly higher-level impressions they prompt, are vivid, irresistible and dependent entirely on basic low-level display parameters (e.g., [4, 5, 19]).

Bingham et al. [57] have provided some empirical support to the hypothesis that bouncing animations can sometimes leave place to perceived animacy, by showing that observers could discriminate real physical bounces from hand-driven bounces with the same period. In our present research we provide a systematic exploration of the conditions by which the visual impression of bouncing may give way to impressions of animate behaviors such as *jumping*. We adopt a rather neutral approach to animacy perception, leaving open the possibility that jumping may not be the only type of animated motion that emerges in bouncing-like scenarios (e.g., the animated motion of an object that goes back-and-forth from a surface).

## Experiment 1

The aim of this first experiment was to provide a broad exploration of animation parameters that lead to the perception of physical motion or animated motion in bouncing-like scenarios. Participants were presented with simple 2D animations such as the one shown in Fig 1, and were asked to indicate if the animation showed a physical bounce, an animated motion, or neither of these two. Four parameters were manipulated. First, the value of *C*, which is a crucial parameter in the physics of bouncing. If observers are sensitive to the violation of conservation laws, as it was suggested in previous studies [37, 38], then we hypothesize that physical bounce impressions should emerge for simulated values of *C* smaller than 1, whereas values greater than 1 would be consistent with the perception of animated motion.

The second manipulated parameter was the *motion pattern*, as the falling object could move with uniform velocity or uniform acceleration (i.e., uniform acceleration in the falling phase and uniform deceleration in the climbing-back phase). This factor has not been manipulated in previous studies on the perception of bouncing, however Vicovaro et al. [58] showed that physically implausible vertical falls characterized by uniform velocity were perceived approximately as natural as accelerated vertical falls that were consistent with terrestrial gravitational acceleration. We wanted to explore whether, also in the case of bouncing, observers are relatively insensitive to the presence or absence of simulated gravitational acceleration.

The third manipulated factor was the *simulated value of gravitational acceleration* (*a*), which could correspond to terrestrial gravitational acceleration or to values smaller than gravitational acceleration. Recent studies have shown that objects are believed to fall vertically downward much more slowly than how they actually fall under the influence of terrestrial gravitational acceleration [59–61]. Based on this prior evidence, we wonder if the perception of physical bounce is more likely under micro-gravity conditions.

Lastly, we also manipulated the *number of bouncing cycles*, which could be one or three.

### Methods

**Transparency and openness.** In this and the following experiments, we report how we determined our sample size, all data exclusions (if any), all manipulations, and all measures in the study, and we follow JARS [62]. Data, samples of stimuli, supplementary figures and supplementary tables are available on OSF at the following link:

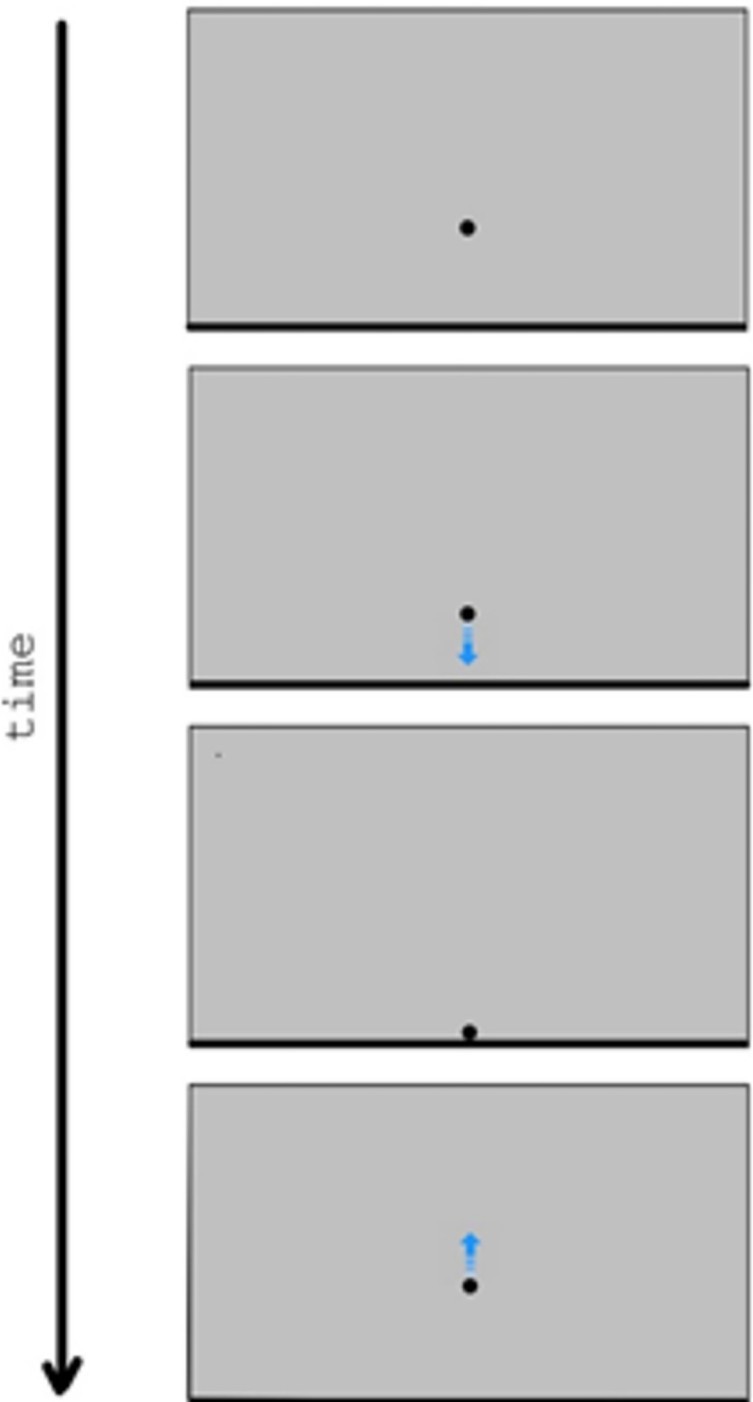

**Fig 1. Example of the animations used in Experiment 1 (four frames).** The blue arrow was added in this figure for illustrative purposes, to show the motion direction of the disk.

https://osf.io/xz8t3/?view_only=02450de951bd48e89e8313a6dff5be52. Data were analyzed using R, version 4.0.4 [63]. This study's design and its analysis were not pre-registered.

**Sample size.** This and the following experiments are exploratory in nature, therefore the sample size is not based on the estimation of a definite expected effect size. We decided to test

30 participants in each experiment, a sample size similar to that of previous studies on visual perception of events [7, 9–11, 13, 17, 64] and larger than the sample size of previous studies on the intuitive physics of bouncing object, in which $N < 15$ [37, 38].

**Participants.** Thirty graduate or undergraduate students at the University of Padova participated in the experiment on a voluntary basis. They were aged from 20 to 31 years (M = 23.63 years, 95% CI [22.59, 24.67]), 18 were women and 12 were men. All participants were naïve to the purpose of the experiment, and gave written informed consent according to the Declaration of Helsinki prior to their inclusion in the experiment. They all reported to have normal or corrected-to-normal visual acuity and were rewarded with 10 €. All procedures used in this and the following experiments were approved by the Ethics Committee for Research in the Human and Social Sciences—CAREUS of the University of Siena (protocol number 194793).

**Stimuli and apparatus.** The stimuli were generated with PsychoPy3 [65] and were presented on a 43.2 cm × 27 cm LCD screen (refresh rate 60 Hz). Participants sat at a distance of about 50 cm from the screen, the background of which was grey [RGB(.506,.506,.506)]. At the beginning of each animation, a black circle (diameter = 1 cm; hereafter *disk*) appeared with its center 4 cm below the center of the screen. A black, thin horizontal rectangle (height = 0.5 cm, length equal to the screen width) also appeared adjacent to the bottom edge of the screen, and remained visible for the whole duration of the animation. This rectangle (hereafter *surface*) was meant to represent a flat horizontal surface against which the disk would bounce at the end of the fall. After 500 ms, the disk started moving vertically downwards according to the parameters described below. A schematic depiction of a stimulus animation is shown in Fig 1.

For convenience, each bouncing cycle can be arbitrarily divided into three phases: the falling phase, the collision, and the climbing-back phase. The *falling phase* lasts from when the disk starts moving downwards to when it collides with the surface. In this experiment, the motion of the disk during the falling phase was given by the combination of factors $a$ and motion pattern. Parameter $a$ could take three different values, that is $g$ (9.81 m/s$^2$), $g/4$ (2.45 m/s$^2$) or $g/16$ (0.61 m/s$^2$). For reference, on the Moon the gravitational acceleration is about $g/6$, on Mars it is about $g/2.6$. The motion pattern could be uniform acceleration or uniform velocity. In the case of uniform acceleration, the disk started moving from rest with acceleration $a$, until it reached a final velocity $v_0 = \sqrt{(2ah_0)}$ just before the collision with the surface ($h_0$ is the initial distance of the disk from the surface, 8.5 cm). As for uniform velocity, the disk moved at a constant velocity $\sqrt{(2ah_0)}/2$. That is, the velocity of the disk corresponded to the average velocity of a disk falling from height $h_0$ with acceleration $a$.

The *collision* is the moment of the impact between the disk and the surface. The crucial parameter is the simulated coefficient of restitution $C$, which determines the disk's post-bounce behavior according to equation $v_1 = -(v_0 \times C)$, where $v_1$ is the velocity of the disk immediately after the collision. Values of $C$ greater than one would be physically implausible because they would imply a violation of energy conservation. In this experiment, $C$ could take nine different values (i.e., 0.25, 0.5, 0.75, 0.875, 1, 1.125, 1.25, 1.5, 1.75). There was no time delay between the moment the front edge of the disk touched the bouncing surface and the beginning of the climbing-back phase. Specifically, the disk made contact with the surface for only one frame.

The *climbing-back phase* lasts from immediately after the collision to when the disk reaches a state of stillness. Similar to the falling phase, the motion of the disk was determined by the combination of factors $a$ and motion pattern. In the case of uniform acceleration, the disk moved with a negative acceleration $-a$ until coming to a stop at height $h_1 = v_1^2/2a$ from the surface. As for uniform velocity, the disk moved at a constant velocity $-\sqrt{(2ah_1)}/2$. That is, the velocity of the disk corresponded to the average velocity of a disk climbing back from the surface to height $h_1$ with acceleration $-a$.

The animations could show either one bouncing cycle or three bouncing cycles. In the case of one bouncing cycle, the animation was stopped immediately after the end of the first climbing-back phase. In the case of the three bouncing cycles, the sequence of falling phase, collision, climbing-back phase was repeated three times (i.e., the animation was stopped immediately after the third climbing-back phase). The falling height at the beginning of each cycle corresponded to the final height of the disk at the end of the previous cycle, as it would be for a real physical bounce.

One-hundred and eight animations were built according to the following factorial design: 2 number of bouncing cycles (1 or 3) × 3 $a$ (0.61 $m/s^2$, 2.45 $m/s^2$, 9.81 $m/s^2$) × 2 motion pattern (uniform acceleration or uniform velocity) × 9 $C$ (0.25, 0.5, 0.75, 0.875, 1, 1.125, 1.25, 1.5, 1.75). In addition, 12 *variable a* animations and 12 *variable C* animations were also created, all showing three bouncing cycles. In the *variable a* animations, a different value of $a$ was used for each bouncing cycle, as if the gravitational attraction varied across bouncing cycles. In these animations, $C$ was kept constant at 0.75. Considering the three bouncing cycles, there were six possible permutations of the three levels of $a$ (e.g., 9.81/2.45/0.61 $m/s^2$, 9.81/0.61/2.45 $m/s^2$, etc.). For each of these permutations, two *variable a* animations were created, one showing uniform acceleration and the other showing the corresponding uniform velocity. Symmetrically, in the *variable C* animations a different value of $C$ was used for each bouncing cycle, whereas $a$ was kept fixed at 2.45 $m/s^2$. Three values of $C$ were selected arbitrarily (i.e., 0.5, 1.0, 1.5), which results again in six possible permutations. There were two animations for each permutation, one showing uniformly accelerated motion and the other showing the corresponding uniform velocity motion.

To sum up, there were a total of 132 animations (i.e., 108 animations from the factorial design + 12 *variable a* animations + 12 *variable C* animations). Each animation was presented three times in a random order, for a total of 396 experimental trials. A sample of the stimuli is available on OSF.

**Procedure.**   Instructions that were readable on the screen informed the participants that they would be presented with bouncing animations, and that their task was to classify each animation into one of three possible categories (i.e., a 3-AFC task). More precisely, participants were presented with the following instructions: "a) *physical bounce* (key 1): the object bounces plausibly against the surface, as if it were an inanimate object (e.g., a ball); b) *animated motion* (key 2): the object moves like a living being with its own internal force; c) *other* (key 3): the animation does not represent a physical bounce or an animated motion." Instructions also invited the participants to trust their first impression. Reviewing the animations was not permitted, and only the numbered keys on the top left of the keyboard could be used to provide the response. We employed a 3-AFC task instead of a 2-AFC task because our initial examination of the stimuli revealed some motions that were not easily described as physical bounce or animated motion. By using a 3-AFC task, we aimed to avoid that participants could classify these ambiguous stimuli as either physical bounce or animated motion, which would have resulted in a greater diversity of visual impressions within the two primary response categories. Instead of using the 3-AFC task, other options could be considered such as using separate rating scales for each of the three response categories, or presenting pairs of stimuli side-by-side and asking participants to choose the stimulus that best represents each category. These methods may provide more detailed information compared to the 3-AFC, especially the rating method. However, it is important to consider that these alternatives may require a longer experimental time and result in a reduction in the number of stimuli used. This may not be ideal since our primary goal was to broadly explore the relationship between the physical parameters and resulting visual impressions.

Each experimental trial started with a 500-ms blank screen, followed by one randomly selected animation. Immediately after the animation ended, a response screen was presented

to remind the participant of the correct key category associations. There were no feedback or time limits for the response. After a response was provided, a new trial started. A custom-duration pause was presented after every 27 experimental trials.

One-hundred sixty-two one bouncing cycle animations (54 animations × 3 repetitions) and 234 three bouncing cycles animations (78 animations × 3 repetitions) were presented in two different experimental sessions that took place at least 24 hours apart, in counterbalanced order. Within each session, the animations showing uniform acceleration and those showing uniform velocity were presented in different blocks, the order of which was counterbalanced across participants. Within each block, the animations were presented in a random order. Before starting each block, participants were presented with randomly selected practice trials to familiarize them with the task (i.e., five practice trials for the one bouncing cycle animations and nine practice trials for the three bouncing cycles animations).

## Results and discussion

Figs 2 and 3 show the mean probabilities, averaged across the participants, of physical bounce responses (blue curves), animated motion responses (red curves), and *other* responses (grey curves) as a function of $C$, separately for uniform acceleration and uniform velocity, and separately for the three values of $a$. Figs 2 and 3 refer respectively to one bouncing cycle and three bouncing cycles animations (the results for the 12 varying $a$ and the 12 varying $C$ animations are not represented in these graphs). An alternative representation of the results is provided in Supplementary Fig S1 and S2 on OSF. These figures show, in each cell, the mean probabilities, averaged across the participants, of physical bounce responses (lower left corner) and animated motion responses (upper right corner) for one bouncing cycle (Fig S1), three bouncing cycles (Fig S2), and varying $a$/varying $C$ animations (Fig S3). Blue cells indicate animations associated with clear impressions of physical bounce, whereas red cells indicate animations associated with clear impressions of animated motion. An animation was classified as an instance of clear physical bounce impression (animated motion impression) if the probability of physical bounce response (animated motion response) exceeded .50. Grey cells correspond to animations that did not elicit clear impressions of physical bounce or animated motion, but that at the same time were associated with a low mean probability of *other* responses (i.e., smaller than .33; please note that the probability of *other* responses is one minus the sum of the other two probabilities). Therefore, grey cells represent somewhat ambiguous animations in between physical bounce and animated motion.

The analyses were conducted on mean response probabilities. In the case of physical bounce, the response was coded as 1 if the participant responded physical bounce and as 0 if the participant responded animated motion or *other*. A similar approach was used for the coding of the other two response categories. The observed probability of each response category was then analyzed using repeated measures ANOVA with Huynh-Feldt correction for the violation of the sphericity assumption. The use of ANOVA for the analysis of dichotomous data has been subject to criticism [66]. However, recent simulation studies [67, 68] have demonstrated that repeated measures ANOVA, with Huynh-Feldt correction, provides better control over Type I and Type II errors than alternative models, such as generalized mixed-effects logit models.

We first run a 2 (Number of bouncing cycles) × 2 (Motion pattern) × 3 ($a$) × 9 ($C$) repeated measures ANOVA on the probability of each response category. These analyses were limited to the data from the 108 animations generated from the main factorial design, whereas the data for the 12 *varying a* and the 12 *varying C* animations were analyzed separately. The results are reported on OSF in Supplementary Tables S1 (physical bounce), S2 (animated motion),

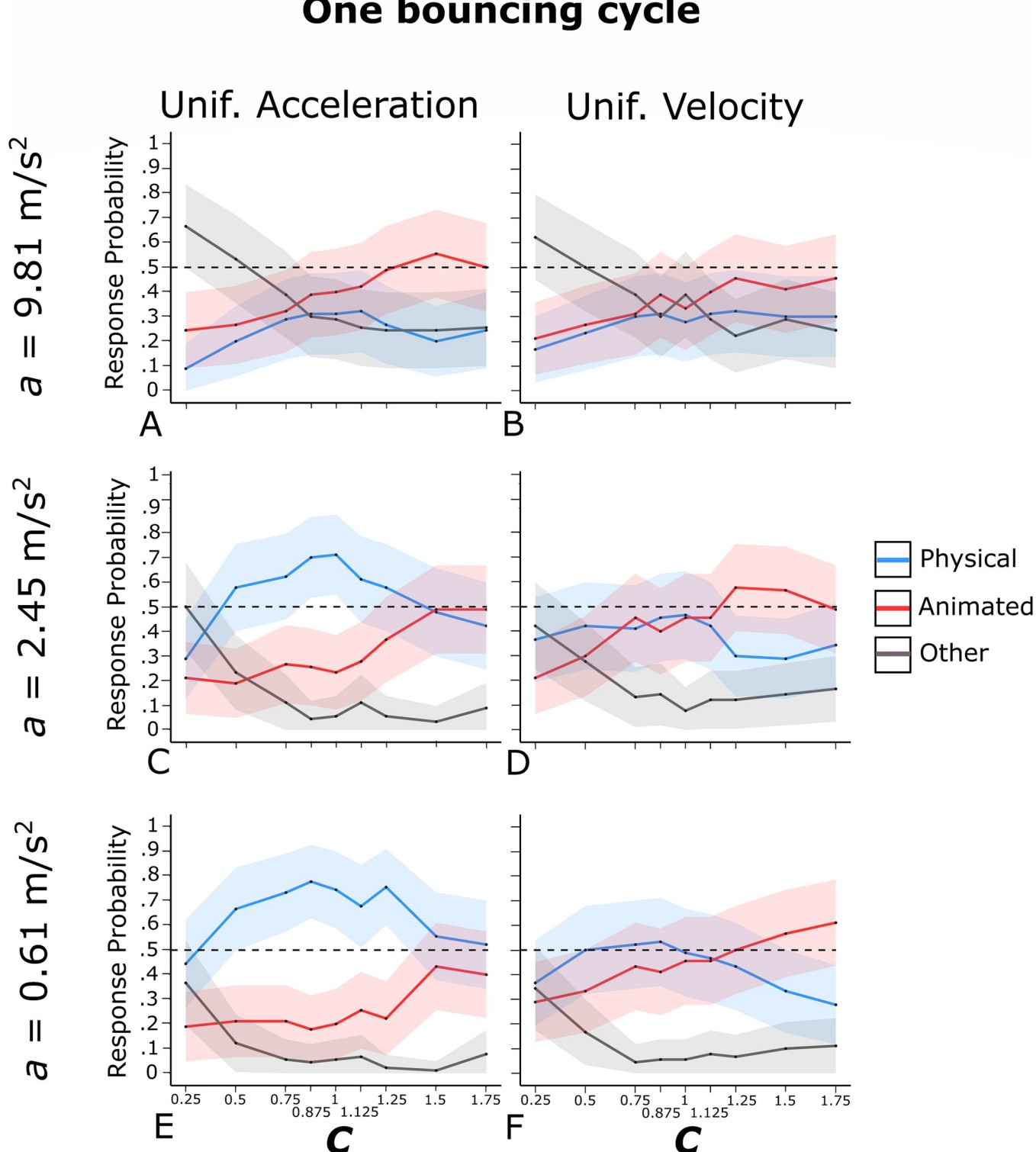

**Fig 2. Observed probability of physical bounce responses (blue curves), animated motion responses (red curves), and *other* responses (grey curves) for the one bouncing cycle animations of Experiment 1.** The shaded area around each curve represents the 95% confidence interval for binomial variables. The horizontal dashed line highlights the .50 response probability threshold.

# Three bouncing cycles

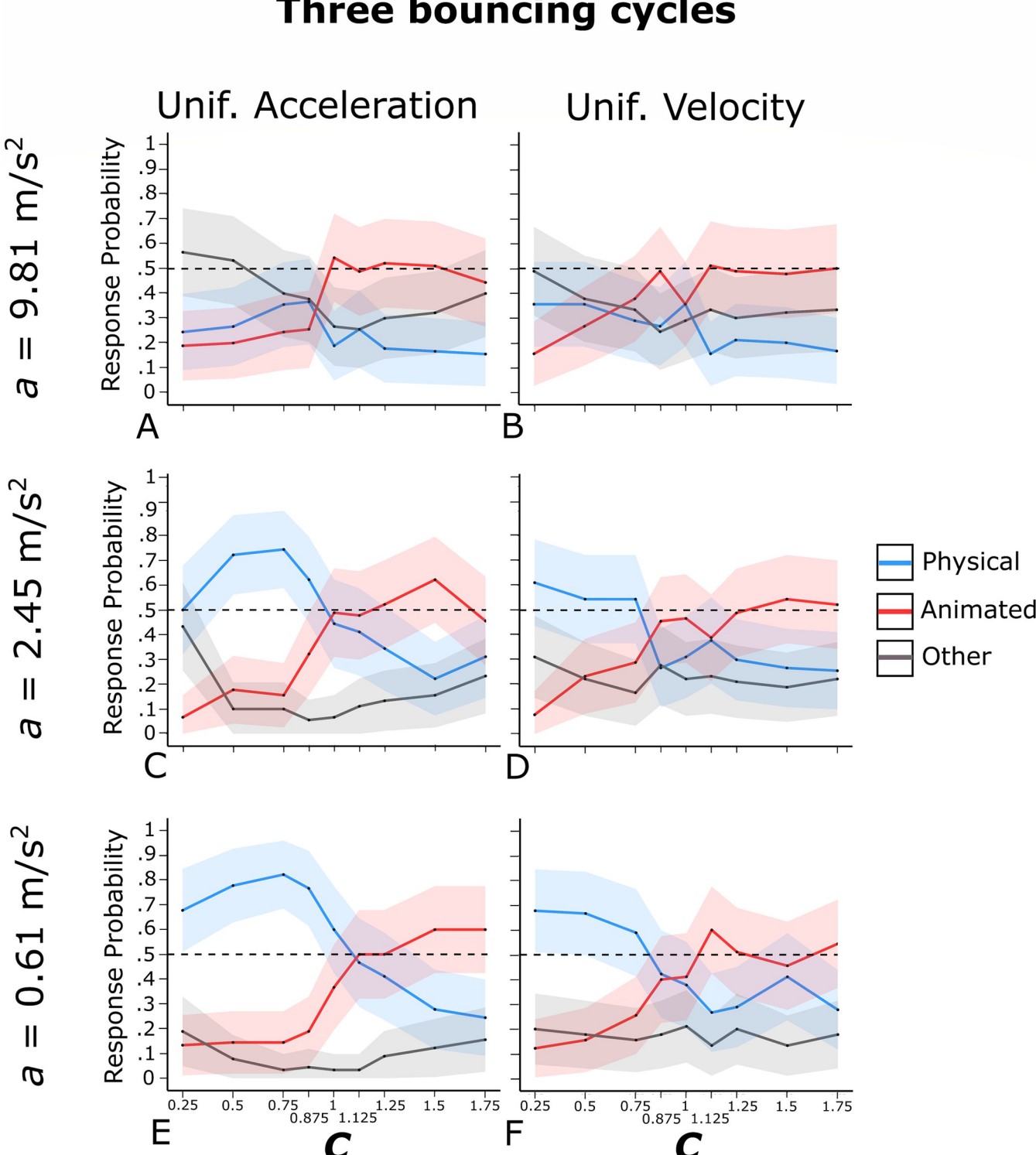

**Fig 3. Observed probability of physical bounce responses (blue curves), animated motion responses (red curves), and *other* responses (grey curves) for the three bouncing cycle animations of Experiment 1.** The shaded area around each curve represents the 95% confidence interval for binomial variables. The horizontal dashed line highlights the .50 response probability threshold.

**Table 1. ANOVA results (Experiment 1) for physical bounce responses, separately for one bouncing cycle and three bouncing cycles.**

| | | | Physical bounce | | | |
|---|---|---|---|---|---|---|
| | | | One bouncing cycle | | | |
| *Predictors* | $df_n$ | $df_d$ | $HF_\varepsilon$ | $F$ | $p$ | $\eta_G^2$ |
| Motion pattern | 1 | 29 | / | 22.09 | **< .001** | .027 |
| *a* | 2 | 58 | 0.641 | 18.55 | **< .001** | .100 |
| *C* | 8 | 232 | 0.376 | 4.68 | **.004** | .041 |
| Motion pattern × *a* | 2 | 58 | 0.938 | 13.71 | **< .001** | .023 |
| Motion pattern × *C* | 8 | 232 | 0.670 | 1.84 | .102 | .007 |
| *a* × *C* | 16 | 464 | 0.537 | 1.78 | .076 | .009 |
| Motion pattern × *a* × *C* | 16 | 464 | 0.948 | 0.89 | .579 | .004 |
| | | | Three bouncing cycles | | | |
| *Predictors* | $df_n$ | $df_d$ | $HF_\varepsilon$ | $F$ | $p$ | $\eta_G^2$ |
| Motion pattern | 1 | 29 | / | 5.65 | **.024** | .008 |
| *a* | 2 | 58 | 0.610 | 27.45 | **< .001** | .082 |
| *C* | 8 | 232 | 0.287 | 9.28 | **< .001** | .104 |
| Motion pattern × *a* | 2 | 58 | 0.737 | 9.65 | **.001** | .007 |
| Motion pattern × *C* | 8 | 232 | 0.832 | 8.14 | **< .001** | .021 |
| *a* × *C* | 16 | 464 | 0.740 | 4.32 | **< .001** | .021 |
| Motion pattern × *a* × *C* | 16 | 464 | 0.899 | 2.35 | **.003** | .009 |

Note: $df_n$ = degrees of freedom at the numerator of the *F* statistic; $df_d$ = degrees of freedom at the denominator of the *F* statistic; $HF_\varepsilon$ = Huynh-Feldt coefficient for the correction of the degrees of freedom (smaller values correspond to stronger violations of the sphericity assumption and lead to stronger corrections of the *p*-value); *F* = value of the *F* statistic; *p* = *p*-value after Huynh-Feldt correction for sphericity violation (boldface type indicates a statistically significant effect); $\eta_G^2$ = generalized eta squared.

and S3 (*other*). The analysis of physical bounce responses revealed a statistically significant four-way interaction. Additionally, there were statistically significant two- and three-way interactions involving the number of bouncing cycles for each of the three response categories. Therefore, to simplify the analysis and discussion of the results, separate ANOVAs were conducted for the one bouncing cycle and the three bouncing cycles conditions. The results are reported in Table 1 (physical bounce) and in Table 2 (animated motion), whereas the results for *other* responses are reported in Supplementary Table S4 on OSF, and they will not be discussed in the main text.

To facilitate the discussion, we will initially focus on the influence of motion pattern, *a* and *C* on physical bounce and animated motion responses, specifically comparing the results for one bouncing cycle and the results for three bouncing cycles. Subsequently, we will address the interaction effects.

**Motion pattern.** As shown in Table 1 and by the comparison between the left-side and the right-side panels in Figs 2 and 3, the probability of physical bounce responses was larger in the case of uniform acceleration than in the case of uniform velocity, in the case of one bouncing cycle (*M* = .49, 95% CI [.31, .66] and *M* = .37, 95% CI [.19, .54]) as well as in the case of three bouncing cycles (*M* = .43, 95% CI [.25, .60] and *M* = .36, 95% CI [.19, .54]). Symmetrically, the probability of animated motion responses was larger in the case of uniform velocity than in the case of uniform acceleration, although the difference reached a statistically significant level in the case of one bouncing cycle (*M* = .42, 95% CI [.24, .59] and *M* = .32, 95% CI [.15, .49]) but not in the case of three bouncing cycles (*M* = .39, 95% CI [.22, .57] and *M* = .37, 95% CI [.19, .54]).

These results suggest that uniform acceleration/deceleration provided a cue to physical bounce, whereas uniform velocity provided a cue to animated motion. However, motion

**Table 2. ANOVA results (Experiment 1) for animated motion responses, separately for one bouncing cycle and three bouncing cycles.**

| Animated motion | | | | | | |
|---|---|---|---|---|---|---|
| One bouncing cycle | | | | | | |
| *Predictors* | $df_n$ | $df_d$ | $HF_\varepsilon$ | $F$ | $p$ | $\eta_G^2$ |
| Motion pattern | 1 | 29 | / | 14.47 | **< .001** | .017 |
| *a* | 2 | 58 | 0.660 | 0.16 | .756 | .001 |
| *C* | 8 | 232 | 0.255 | 6.77 | **.002** | .060 |
| Motion pattern × *a* | 2 | 58 | 0.844 | 12.97 | **< .001** | .019 |
| Motion pattern × *C* | 8 | 232 | 0.705 | 1.36 | .236 | .005 |
| *a* × *C* | 16 | 464 | 0.744 | 0.91 | .534 | .005 |
| Motion pattern × *a* × *C* | 16 | 464 | 1.045 | 0.83 | .650 | .003 |
| Three bouncing cycles | | | | | | |
| *Predictors* | $df_n$ | $df_d$ | $HF_\varepsilon$ | $F$ | $p$ | $\eta_G^2$ |
| Motion pattern | 1 | 29 | / | 0.84 | .367 | .001 |
| *a* | 2 | 58 | 0.552 | 0.15 | .726 | < .001 |
| *C* | 8 | 232 | 0.279 | 13.58 | **< .001** | .139 |
| Motion pattern × *a* | 2 | 58 | 0.878 | 0.05 | .931 | < .001 |
| Motion pattern × *C* | 8 | 232 | 0.865 | 5.19 | **< .001** | .013 |
| *a* × *C* | 16 | 464 | 0.675 | 2.40 | **.007** | .011 |
| Motion pattern × *a* × *C* | 16 | 464 | 0.906 | 1.48 | .110 | .005 |

Note: $df_n$ = degrees of freedom at the numerator of the *F* statistic; $df_d$ = degrees of freedom at the denominator of the *F* statistic; $HF_\varepsilon$ = Huynh-Feldt coefficient for the correction of the degrees of freedom (smaller values correspond to stronger violations of the sphericity assumption and lead to stronger corrections of the *p*-value); *F* = value of the *F* statistic; *p* = *p*-value after Huynh-Feldt correction for sphericity violation (boldface type indicates a statistically significant effect); $\eta_G^2$ = generalized eta squared.

pattern had a stronger effect in the case of one bouncing cycle ($\eta_G^2$ = .027 for physical bounce responses and $\eta_G^2$ = .017 for animated motion responses) than in the case of three bouncing cycles ($\eta_G^2$ = .008 for physical bounce responses and $\eta_G^2$ = .001 for animated motion responses). Fig S1 shows that, in the case of one bouncing cycle, 14 out of the 16 animations that gave rise to clear physical bounce impressions were characterized by uniform acceleration (87.5%), whereas 4 out of 5 clear animated motion impressions were characterized by uniform velocity (80%). These percentages were comparatively lower in the case of three bouncing cycles (i.e., 57.1% for physical bounce and 46.1% for animated motion, see Fig S2).

*a.* The probability of physical bounce responses decreased with *a* (9.81 m/s²: *M* = .26, 95% CI [.10, .41]; 2.45 m/s²: *M* = .45, 95% CI [.27, .63]; 0.61 m/s²: *M* = .52, 95% CI [.34, .70]). No clear differences between one and three bouncing cycles emerged, as also demonstrated by the large *p*-value and the negligible effect size for the interaction between the number of bouncing cycles and *a* in the four-way ANOVA (*p* = .654 and $\eta_G^2$ = < .001, see Supplementary Table S1). As for animated motion impressions, *a* had a negligible non-significant effect, again with no apparent differences between one and three bouncing cycles (*p* = .910 and $\eta_G^2$ = < .001 for the interaction between the number of bouncing cycles and *a* in the first four-way ANOVA, see Supplementary Table S2).

Surprisingly, none of the animations characterized by *a* = 9.81 m/s² (i.e., the terrestrial gravitational acceleration) gave rise to a clear impression of physical bounce (see Figs 2, 3, S1 and S2). That is, not only physical bounce impressions were favored by simulated micro-gravity, but simulated terrestrial gravitational attraction was actually incompatible with physical bounce impressions. We will return on this important result in the General Discussion.

**C.** Parameter *C* had a significant effect on physical bounce responses in the case of one bouncing cycle as well as in the case of three bouncing cycles. The effect size was noticeably larger for three bouncing cycles ($\eta_G^2$ = .104) compared to one bouncing cycle ($\eta_G^2$ = .041). Furthermore, the impact of *C* on physical bounce responses differed considerably between one bouncing cycle (Fig 2) and three bouncing cycles (Fig 3) at a qualitative level.

In Fig 3 (three bouncing cycles), the blue curves for physical bounce responses were similar in shape to classic psychometric curves. Except when *a* = 9.81 m/s², relatively large probability values emerged for values of *C* in the range from 0.25 to 0.75, then probability decreased steeply in the range from 0.75 to 1.5 (see Fig 3C–3F). This suggests that observers were very sensitive to the physical plausibility of *C* in the case of three bouncing cycles, which is also supported by the fact that clear physical bounce impressions emerged only when *C* was physically plausible (i.e., $C \leq 1$) whereas clear animated motion impressions emerged only when *C* was physically implausible (i.e., $C > 1$; see Fig 3 and S2). As shown in Fig 3C–3F, the intersection between physical bounce and animated motion responses occurred in the interval from 0.875 to 1.125 (i.e., close to 1, which is the physical threshold of plausibility of *C*).

In Fig 2 (one bouncing cycle), the blue curves for physical bounce responses showed a reversed U-shape pattern with an initially increasing trend, a peak for values around 0.875–1.25, and then a decreasing trend. As shown in Fig 2E and S1, with uniform acceleration and *a* = 0.61 m/s², clear physical bounce impressions emerged for all values of *C* except 0.25. Therefore, even animations showing extreme violations of energy conservation (e.g., *C* = 1.75) led to the perception of clear physical bounce. In other words, provided that uniform acceleration was present, almost all values of *C* elicited a clear physical bounce impression. Additionally, it is noteworthy that, with uniform acceleration and *a* = 2.45 m/s² or *a* = 0.61 m/s² (Fig 2C and 2E), physical bounce responses were always more likely than animated motion responses for all values of *C*. Indeed, there was no intersection between physical bounce responses and animated motion responses in these cases. Surprisingly enough, the only value of *C* associated with a low probability of physical bounce responses was a physically plausible one (i.e., 0.25). This can be explained by the fact that, in the case of one bouncing cycle with *C* = 0.25, the climbing-back phase was very short, therefore the animation ended rather abruptly after the first bounce (i.e., the animation was probably too short to provide a clear bouncing impression; this hypothesis is also supported by the high observed probability of *other* responses).

As for animated motion impressions, they tended to increase with *C*, in the case of one bouncing cycle as well as in the case of three bouncing cycles (see Table 2, Figs 2 and 3). Again, the effect was clearly more pronounced in the case of three bouncing cycles ($\eta_G^2$ = .139) than in the case of one bouncing cycle ($\eta_G^2$ = .060; see also the significant interaction between the number of bouncing cycles and *C* in Table S2).

To sum up, the results of these analyses indicate that the visual system is attuned to the physical plausibility of *C* in the case of three bouncing cycles, but not in the case of one bouncing cycle. It is also worth noting that the opposite tended to be true for the motion pattern. In other words, the number of bouncing cycles modulates the influence of *C* and motion pattern on the participants' responses. The theoretical implications of these results will be discussed in more details in the General Discussion.

**Interaction effects.** Table 1 highlights that for physical bounce impressions with three bouncing cycles, all two-way interactions and the three-way interaction were found to be statistically significant. In terms of interpreting these interactions, the visual inspection of Fig 3 suggests that the effect of each factor on the probability of physical bounce responses was more pronounced at the level(s) of the other factor(s) that were related to a higher likelihood of physical bounce impressions. For instance, the effect of *C* was more prominent in the case of uniform acceleration as compared to uniform velocity. Additionally, it was stronger for the

two smaller values of *a* in comparison to the largest value. The interactions observed for physical bounce impressions with one bouncing cycle (Table 1, Fig 2), as well as for the statistically significant interactions seen for animated motion impressions (Table 2, Figs 2 and 3), seem to follow a similar pattern.

**Results for the "varying *a*" and the "varying *C*" animations.** As shown in Fig S3, the *varying a* animations were mostly associated with low probabilities of physical bounce responses (*M* = .24, 95% CI [.16, .32]) and with moderate probabilities of animated motion (*M* = .37, 95% CI [.25, .48]) and *other* responses (*M* = .40, 95% CI [.28, .51]). None of these animations gave rise to clear physical bounce or animated motion impressions.

The *varying C* animations were mostly associated with low probabilities of physical bounce responses (*M* = .24, 95% CI [.16, .31]), low probabilities of *other* responses (*M* = .31, 95% CI [.21, .41]), and relatively high probabilities of animated motion responses (*M* = .46, 95% CI [.36, .55]). Only two of these animations gave rise to clear animated motion impressions, however a probability close to .50 was observed for most of these animations. Fig 4 shows that the only animation associated with a relatively low probability of animated motion response was the one in which *C* decreased across the three bouncing cycles.

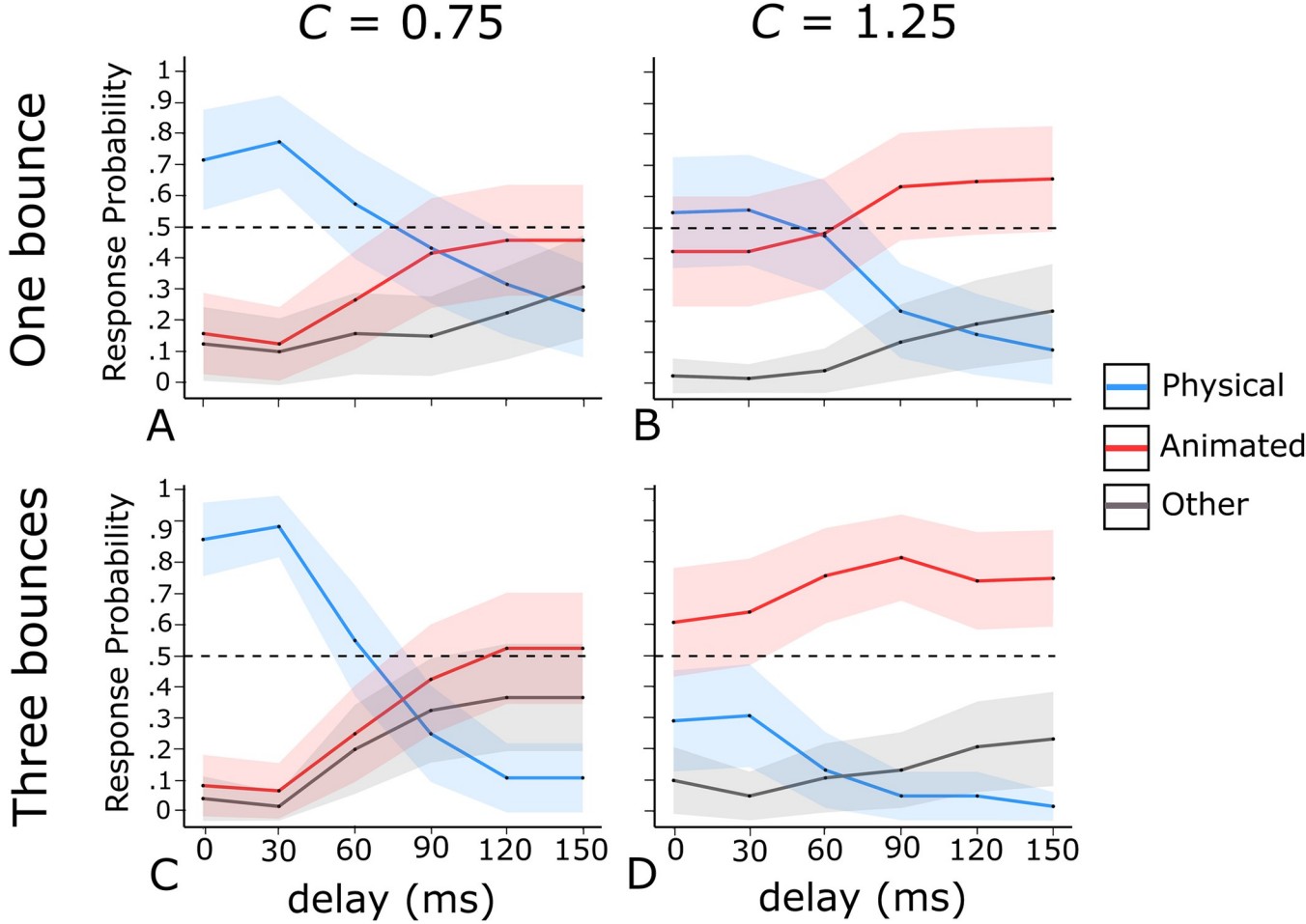

**Fig 4. Observed probability of physical bounce responses (blue curves), animated motion responses (red curves), and *other* responses (grey curves) for the four animations types of Experiment 2, as a function of the delay.** The shaded area around each curve represents the 95% confidence interval for binomial variables. The horizontal dashed line highlights the .50 response probability threshold. Panels A and B: results for one bouncing cycle animations. Panels C and D: results for the three bouncing cycles animations. Panels A and C: results for *C* = 0.75. Panels B and D: results for *C* = 1.25.

## Experiment 2

In the second experiment we tested to see if and how a short delay at the moment of the collision would affect visual impressions of physical bounce and animated motion. Michotte [2] showed that observers could perceive a clear launching effect when a 0–70 ms delay was present between the arrival of *A* and the departure of *B*. The strength of the launching effect was even improved by a 30–40 ms delay, whereas delays longer than 70 ms tended to progressively weaken and then to disrupt the launching impression (see also [1], for a review of studies on the influence of timing on phenomenal causality). Michotte [2] argued that the reason why short delays do not disrupt the launching effect despite their physical implausibility is that such delays do preserve the perceived temporal continuity of the motions of *A* and *B*, which is necessary for the effect to occur.

From a physical viewpoint, the delay between the moment in which a falling object touches the bouncing surface and the subsequent climbing-back phase is typically negligible (e.g., 5 ms for a tennis ball, 0.1 ms for a steel ball [32]). While the delay might be longer for certain objects, like a rubber ball that is not fully inflated, it is noteworthy that the stimuli utilized in our experiments did not display any noticeable deformation upon impact with the bouncing surface. The stimuli resembled rigid steel balls more than soft rubber balls, as seen in the stimuli provided on OSF. Therefore, if the visual system is sensitive to the plausibility of the event's physical properties, delays as brief as 30 milliseconds should be enough to disrupt the impression of a physical bounce. Alternatively, if the key factor is the perceived temporal continuity between the falling and the climbing-back phase, then the tolerance to delays should be similar to that reported by Michotte [2] in the case of the launching effect (i.e., 0–70 ms).

As for the relationship between temporal delays and animated motion, the behavior of living bodies can be compared to that of a spring, especially when one considers the biomechanics of jumping. The body as a whole shrinks at the moment of the contact with the surface and then expands to bounce off the surface. This shrinking-expansion process plausibly takes a relatively long time, therefore we speculate that relatively long delays between the contact with the surface and the climbing-back phase may favor visual impressions of animated motion (i.e., jumping).

### Methods

**Participants.**   Thirty graduate or undergraduate students at the University IUAV of Venice participated in the experiment on a voluntary basis. They were aged from 20 to 31 years (M = 24.4 years, 95% CI [23.3, 25.5]), 17 were women and 13 were men. All participants were naïve to the purpose of the experiment, and gave written informed consent according to the Declaration of Helsinki prior to their inclusion in the experiment. They all reported normal or corrected-to-normal visual acuity and were rewarded with 10 €. None of them had taken part in Experiment 1.

**Stimuli and apparatus.**   The stimuli were similar to those used in Experiment 1 and were presented on a 25.5 cm × 14.5 cm screen (refresh rate 60 Hz). Participants sat at a distance of about 50 cm from the screen.

There were four different *animation types*, resulting from a 2 number of bouncing cycles (1 or 3) × 2 *C* (0.75 or 1.25) factorial design. Based on the results of Experiment 1, which showed that *g*/4 was a suitable acceleration value for creating clear visual impressions of physical bounce and animated motion, all animations in Experiment 2 were produced with a uniform acceleration of *a* = *g*/4. According to the results of Experiment 1, when no delay is present, *C* = 0.75 should give rise to clear visual impressions of physical bounce both in the case of one bouncing cycle (Fig 2 and Fig S1) and as in the case of three bouncing cycles (Fig 3 and Fig

S2). In the case of one bouncing cycle, $C = 1.25$ should be associated with high probability of physical bounce responses but also with non-negligible probability of animated motion responses (i.e., the resulting impressions should be quite ambiguous, see Fig 2 and Fig S1). In the case of three bouncing cycles, $C = 1.25$ should give rise to clear visual impressions of animated motion (Fig 3 and Fig S2).

We explored if and how these impressions were modified by the addition of a delay at the moment of the contact of the disk with the surface. There was a total of 96 experimental trials, given by the following design: 4 animation type × 6 delay (i.e., 0, 30, 60, 90, 120, or 150 ms) × 4 repetitions. The stimuli are available on OSF.

**Procedure.** Everything was identical to Experiment 1, with the following exceptions. The experiment took place in a single session. The 48 one bouncing cycle animations and the 48 three bouncing cycles animations were presented in different blocks. The order of the blocks was counterbalanced across participants. Within each block the stimuli were presented in random order. Each experimental block was preceded by 12 randomly selected practice trials.

## Results and discussion

For each of the four animation types, Fig 4 shows the probability of physical bounce response, animated motion response, and *other* response as a function of the delay, averaged across the participants.

Initially, we conducted a repeated measures ANOVA with Huynh-Feldt correction on the probability of each response category, using a 2 (Number of bouncing cycles) × 2 (C) × 6 (delay) design. The purpose of this analysis was to determine whether the effect of delay on the probability of each response category was modulated by the animation type. We present the results for physical bounce and animated motion responses in Table 3, whereas the results for *other* responses are reported in Supplementary Table S7.

To investigate the impact of delay on the probability of physical bounce and animated motion responses for each animation type, we then conducted one-way repeated measures ANOVAs with Huynh-Feldt correction, separately for each animation type. The results are summarized in Table 4. We further examined the findings using post-hoc comparisons with Bonferroni correction (see Supplementary Table S5 on OSF for physical bounce responses and Supplementary Table S6 on OSF for animated motion responses). All the results for *other* responses are summarized in Supplementary Table S7.

Finally, we compared the probability of physical bounce and animated motion responses at each delay level for the four animation types, using paired-sample *t*-tests with Bonferroni correction. We present the results of this analysis in Table 5.

In the following two subsections, we will discuss the main findings regarding physical bounce impressions and animated motion impressions separately. In a third subsection, we will explore the direct comparisons between the two types of impressions. Supplementary Table S7 on OSF provides the results for *other* responses.

**Effects of delay on physical bounce impressions.** Regarding physical bounce impressions, we found a significant three-way interaction (Table 3), indicating that the impact of delay on response probabilities differed depending on the combination of number of bouncing cycles and $C$ (i.e, depending on the animation type). One-way ANOVAs with Huynh-Feldt correction (Table 4) showed a significant effect of delay for all animation types, with differences in the strength of this effect. Notably, the four animation types had distinct response probability values at the 0 ms delay, ranging from low values for three bouncing cycles with $C = 1.25$ (Fig 4D) to high values for three bouncing cycles with $C = 0.75$ (Fig 4C). Consequently, the variability in physical bounce response probabilities was lower for the former than

**Table 3. ANOVA results (Experiment 2) for the effects of number of bouncing cycles, *C*, and delay on physical bounce and animated motion responses.**

| | | Physical bounce | | | | |
|---|---|---|---|---|---|---|
| Predictors | $df_n$ | $df_d$ | $HF_\varepsilon$ | F | p | $\eta_G^2$ |
| N_bounces | 1 | 29 | / | 41.11 | < **.001** | .043 |
| C | 1 | 29 | / | 20.14 | < **.001** | .149 |
| Delay | 5 | 145 | 0.384 | 67.98 | < **.001** | .337 |
| N_bounces × C | 1 | 29 | / | 5.15 | **.031** | .020 |
| N_bounces × Delay | 5 | 145 | 0.833 | 2.61 | **.037** | .009 |
| C × Delay | 5 | 145 | 0.841 | 11.14 | < **.001** | .040 |
| N_bounces × C × Delay | 5 | 145 | 0.714 | 11.31 | < **.001** | .033 |
| | | Animated motion | | | | |
| Predictors | $df_n$ | $df_d$ | $HF_\varepsilon$ | F | p | $\eta_G^2$ |
| N_bounces | 1 | 29 | / | 8.86 | **.006** | .015 |
| C | 1 | 29 | / | 24.87 | < **.001** | .178 |
| Delay | 5 | 145 | 0.270 | 12.80 | < **.001** | .113 |
| N_bounces × C | 1 | 29 | / | 6.15 | **.019** | .016 |
| N_bounces × Delay | 5 | 145 | 0.752 | 0.38 | .814 | .001 |
| C × Delay | 5 | 145 | 0.784 | 6.17 | **.002** | .016 |
| N_bounces × C × Delay | 5 | 145 | 0.769 | 2.37 | .059 | .007 |

**Note**: $df_n$ = degrees of freedom at the numerator of the *F* statistic; $df_d$ = degrees of freedom at the denominator of the *F* statistic; $HF_\varepsilon$ = Huynh-Feldt coefficient for the correction of the degrees of freedom (smaller values correspond to stronger violations of the sphericity assumption and lead to stronger corrections of the *p*-value); *F* = value of the *F* statistic; *p* = *p*-value after Huynh-Feldt correction for sphericity violation (boldface type indicates a statistically significant effect); $\eta_G^2$ = generalized eta squared.

the latter case, which could account for the differing magnitudes of the delay effect across the four animation types (Fig 4D).

Upon visually inspecting Fig 4 and conducting post-hoc *t*-tests with Bonferroni correction (Supplementary Table S5), we observed that despite differences in magnitude, delay had

**Table 4. ANOVA results (Experiment 2) for the effects of delay on the probability of physical bounce and animated motion responses, separately for each combination of number of bouncing cycles and *C*.**

| | | Physical bounce | | | | |
|---|---|---|---|---|---|---|
| Animation types | $df_n$ | $df_d$ | $HF_\varepsilon$ | F | p | $\eta_G^2$ |
| One bounce, C = 0.75 | 5 | 145 | 0.764 | 21.52 | < **.001** | .272 |
| One bounce, C = 1.25 | 5 | 145 | 0.498 | 21.06 | < **.001** | .244 |
| Three bounces, C = 0.75 | 5 | 145 | 0.752 | 94.20 | < **.001** | .637 |
| Three bounces, C = 1.25 | 5 | 145 | 0.414 | 10.33 | < **.001** | .182 |
| | | Animated motion | | | | |
| Animation types | $df_n$ | $df_d$ | $HF_\varepsilon$ | F | p | $\eta_G^2$ |
| One bounce, C = 0.75 | 5 | 145 | 0.573 | 11.23 | < **.001** | .157 |
| One bounce, C = 1.25 | 5 | 145 | 0.432 | 4.08 | **.002** | .074 |
| Three bounces, C = 0.75 | 5 | 145 | 0.407 | 20.77 | < **.001** | .250 |
| Three bounces, C = 1.25 | 5 | 145 | 0.376 | 1.99 | .148 | .039 |

**Note**: $df_n$ = degrees of freedom at the numerator of the *F* statistic; $df_d$ = degrees of freedom at the denominator of the *F* statistic; $HF_\varepsilon$ = Huynh-Feldt coefficient for the correction of the degrees of freedom (smaller values correspond to stronger violations of the sphericity assumption and lead to stronger corrections of the *p*-value); *F* = value of the *F* statistic; *p* = *p*-value after Huynh-Feldt correction for sphericity violation (boldface type indicates a statistically significant effect); $\eta_G^2$ = generalized eta squared.

similar qualitative effects on the probability of physical bounce responses across the four animation types. Specifically, a 30 ms delay resulted in a non-significant increase in the probability of physical bounce responses compared to a 0 ms delay. The probability of physical bounce responses started decreasing steeply in the passage from 30 ms to 60 ms (except for one bouncing cycle with $C = 1.25$, where the decrease was observed between 60 and 90 ms, as seen in Fig 4B). Post-hoc tests indicated that for all four animation types, the probability of physical bounce responses was significantly higher at 0 ms and 30 ms compared to 90 ms, 120 ms, and 150 ms. Comparisons between 0 ms and 30 ms versus 60 ms yielded mixed results. The difference between 0 ms and 60 ms was statistically significant only for three bouncing cycles with $C = 0.75$, while the difference between 30 ms and 60 ms was statistically significant for both three bouncing cycles and one bouncing cycle with $C = 0.75$.

**Effects of delay on animated motion impressions.** The pattern of results for animated motion responses mirrored that of physical bounce responses in a substantially symmetrical manner. Specifically, the probability of animated motion responses decreased slightly from 0 ms to 30 ms, followed by a steep increase from 30 ms onwards. However, the magnitude of the effect of delay varied across the four animation types, as evidenced by Table 4 and the results of post-hoc comparisons (Supplementary Table S6). The reasons for these variations are similar to those discussed earlier in reference to physical bounce responses.

It is important to note that the effects of delay on animated motion impressions were smaller overall in comparison to the effects on physical bounce impressions. This can be seen from the fact that the three-way interaction was not statistically significant for animated motion (Table 3), and the effects of delay on animated motion did not reach a statistically significant level for three bouncing cycles and $C = 1.25$ (Table 4). Upon visual inspection of Fig 4, it appears that this may be due to the fact that longer delays of 120 ms and 150 ms were not only associated with a high probability of animated motion responses, but also with a relatively high probability of *other* responses (as seen in the analyses of *other* responses in Supplementary Table S7). In other words, while these longer delays did provide cues to animated motion, the resulting impression may have been somewhat ambiguous, especially when the value of $C$ tended to favor visual impressions of physical bounce (i.e., $C = 0.75$, as seen in Fig 4A and 4C).

**Direct comparisons between physical bounce and animated motion responses at each delay level.** In addition to the previous analyses, we conducted paired-sample $t$-tests with Bonferroni correction to compare the probability of physical bounce and animated motion responses at each delay level, for each animation type. Table 5 summarizes the results of these tests. When $C$ favored visual impressions of physical bounce ($C = 0.75$), physical bounce responses were more frequent than animated motion responses for delay values ranging from

**Table 5. Results for paired sample $t$-tests with Bonferroni correction on the mean probability of physical bounce responses and the mean probability of animated motion responses, separately for the four animation types.**

|  | One bounce, $C = 0.75$ | | Three bounces, $C = 0.75$ | | One bounce, $C = 1.25$ | | Three bounces, $C = 1.25$ | |
|---|---|---|---|---|---|---|---|---|
|  | $t(29)$ | $p_{bonf}$ | $t(29)$ | $p_{bonf}$ | $t(29)$ | $p_{bonf}$ | $t(29)$ | $p_{bonf}$ |
| **0 ms** | 5.50 | $< .001$ | 12.24 | $< .001$ | 0.84 | .999 | -2.25 | .194 |
| **30 ms** | 6.97 | $< .001$ | 13.26 | $< .001$ | 0.87 | .999 | -2.14 | .240 |
| **60 ms** | 3.20 | **.020** | 3.33 | **.014** | -0.68 | .999 | -7.88 | $< .001$ |
| **90 ms** | 0.14 | .999 | -1.59 | .734 | -4.00 | **.002** | -12.32 | $< .001$ |
| **120 ms** | -1.19 | .999 | -3.83 | **.004** | -4.85 | $< .001$ | -8.83 | $< .001$ |
| **150 ms** | -1.93 | .378 | -3.85 | **.004** | -6.22 | $< .001$ | -10.21 | $< .001$ |

**Note**: Positive values of $t$ indicate that the probability of physical bounce responses exceeded the probability of animated motion responses.

0 ms to 60 ms. At a delay of 90 ms, there was no significant difference between the two impressions, while for delays of 120 ms and 150 ms, animated motion responses were more frequent than physical bounce responses. However, when $C$ favoured visual impressions of animated motion ($C = 1.25$ with three bouncing cycles) or an ambiguous visual impression ($C = 1.25$ with one bouncing cycle), delays of 60 ms or longer led to a significant prevalence of animated motion responses over physical bounce responses.

To sum up, the pattern of results is strikingly reminiscent of the results reported by Michotte [2] about the effects of delay on the launching effect. He found that a short 30–40 ms delay between the arrival of $A$ and the departure of $B$ could strengthen the launching impression; in a similar vein, we found that, with respect to no delay, a 30 ms delay led to a slight increase in the probability of physical bounce impressions for all four animation types, and to a corresponding decrease in the probability of animated motion impressions. Michotte [2] also found that the launching effect was still present with a 60–70 ms delay, and similarly we found that physical bounce impressions still prevailed over animated motion impressions with a 60 ms delay (with $C = 0.75$). Lastly, Michotte had found that with delays longer than 60–70 ms, the visual impressions of launching left place to the impression that the motion of $B$ was not passively induced by the collision with $A$, but rather looked active (i.e., self-generated). In a similar vein we found that, with delays of 90 ms or longer, physical bounce impressions tend to give way to animated motion impressions.

According to Michotte [2], short delays do not disrupt the launching effect because, despite their being physically implausible, they preserve the perceived temporal contiguity of the motions of $A$ and $B$. In line with this idea we suggest that delays up to 60–70 ms do not disrupt physical bounce impressions because they preserve the perceived temporal contiguity between the falling phase and the climbing back phase. This suggest that the necessary condition for the perception of physical bounce would be the perceived temporal continuity between the falling and the climbing-back phase, above and beyond physical realism.

As expected, the presence of a temporal discontinuity between the falling and the climbing-back phase provides a cue to animated motion. However, it is important to note that even with long delays, there was no significant difference between the probability of physical bounce responses and animated motion responses in the case of one bouncing cycle with $C = 0.75$. Thus, the results for animations with $C = 0.75$ suggest that the delay alone is not sufficient to create strong impressions of animated motion. The combination of a long delay and $C > 1$ is necessary to produce convincing animated motion impressions (see Fig 4B and 4D).

## Experiment 3

In Experiments 1 and 2 as well as in previous studies on bouncing perception, participants were presented with scenarios showing a visible opaque bouncing surface. In this third experiment we tested to see if the presence of an opaque visible bouncing surface is actually necessary for the visual perception of physical bounce and animated motion.

Our hypothesis is that, when the kinematics of the disk is perceptually consistent with a physical bounce or an animated jump, the visual system might automatically *create* a transparent bouncing surface against which the object bounces or jumps. This hypothesis is corroborated by the results of a recent study by Little and Firestone [69], in which participants were presented with a disk that fell vertically downward and then exited either with a vertical trajectory (as in our work) or with an oblique trajectory. Results showed that the vertical trajectory was associated to the perception of a horizontal amodal surface, whereas the oblique trajectory was associated to the perception of an amodal oblique surface. The authors concluded that the visual system can infer postdictively the presence of an amodal bouncing surface oriented

coherently with the object's trajectory. More generally, it has been suggested that when the visual context indicates the possible presence of a meaningful relationship between the objects in the scene, but one of the objects involved in the relation is actually missing, the visual system can fill in the missing element [4, 17].

In Experiment 3, we tested to see if and how the removal of the visible opaque surface affected the perception of bouncing and jumping. Based on the results of Experiment 1, we selected animations from the most prototypical ones for each response category (i.e., physical bounce, animated motion, *other*). The black horizontal bar representing the bouncing surface was removed from the scene (see Fig 5), and we compared the relative probabilities of the three response categories across Experiments 1 and 3.

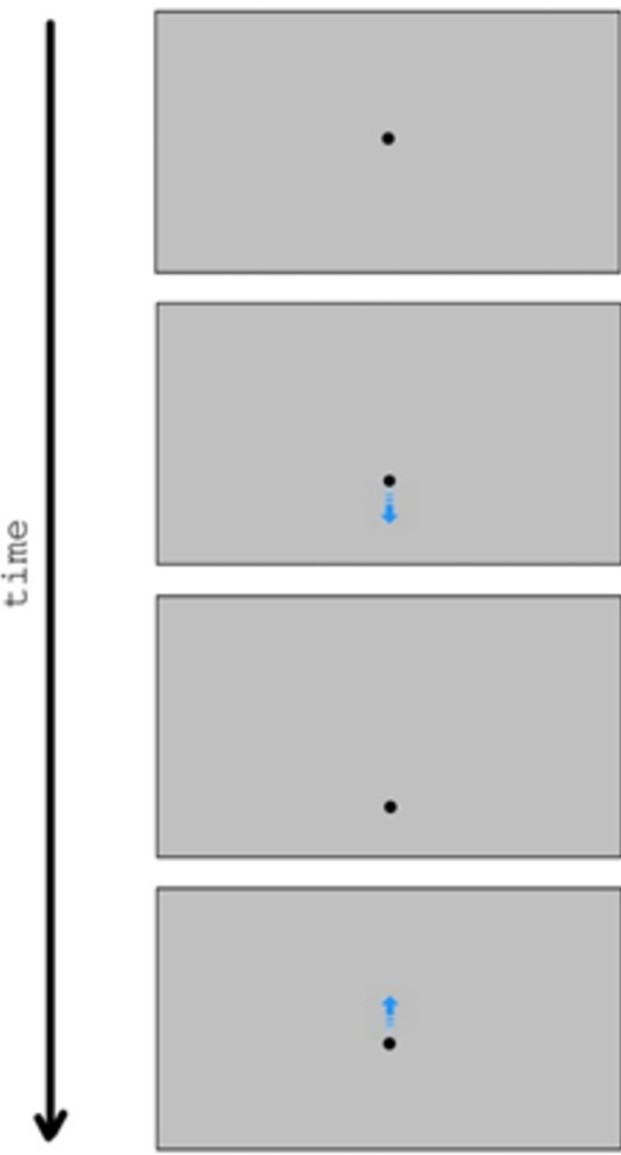

**Fig 5. Example of the animations used in Experiment 3 (four frames).** The blue arrow was added in this figure for illustrative purposes, to show the motion direction of the disk.

## Methods

**Participants.** Thirty graduate or undergraduate students at the University of Padova participated in the experiment on a voluntary basis. They were aged from 19 to 33 years ($M$ = 22.97 years, 95% CI [22.06, 23.87]), 24 were women and 6 were men. All participants were naïve to the purpose of the experiment, and gave written informed consent according to the Declaration of Helsinki prior to their inclusion in the experiment. They all reported normal or corrected-to-normal visual acuity. None of them had participated in Experiment 1 or Experiment 2. At the end of the experiment they were rewarded with 10 €.

**Stimuli and apparatus.** Everything was identical to Experiment 1, with the following exceptions. First, there was no horizontal bar adjacent to the bottom edge of the screen; second, at the beginning of each animation the disk appeared at the center of the screen, rather than 4 cm below it. The total distance travelled by the disk during the falling phase was 8.5 cm, as in Experiment 1. Therefore, the point of direction reversal of the disk was 5 cm above the bottom edge of the screen (considering the center of the disk), rather than just 1 cm above. There was a clearly visible gap between the point of reversal and the bottom edge of the screen, which prevented that the latter could serve as a visible bouncing surface. A schematic depiction of a stimulus animation is shown in Fig 5.

There were 24 different animations, defined according to the following criteria. For each response category (i.e., physical bounce, animated motion, *other*), we selected four animations that, according to the results of Experiment 1, were judged as clearly belonging to that category. This was done separately for the one bouncing cycle animations and the three bouncing cycles animations [i.e., 2 number of bouncing cycles (1 or 3) × 3 response categories (physical bounce, animated motion, *other*) × 4 animations].

The parameters corresponding to the selected animations are reported in Fig 6, together with the probabilities of physical bounce and animated motion responses from Experiment 1 (first row in each pair of rows). For consistency, all the prototypical animations showed uniform acceleration as motion pattern. The stimuli are available on OSF.

**Procedure.** The procedure was the same as in Experiment 1, with the following exceptions. Participants were presented with these instructions: "a) physical bounce on an invisible surface (key 1): the motion direction change of the disk appears to be due to the collision against an invisible surface, as if the disk bounced against an invisible surface; b) jump on an invisible surface (key 2): the disk appears to jump on an invisible surface, as if it were a living being with its own internal force; c) *other* (key 3): the motion direction change of the disk does not appear to be due to a physical bounce or to a jump against and invisible surface." Note that "jump on an invisible surface" was used instead of "animated motion" because all the prototypical stimuli of animated motion appeared to be instances of jumping. Each animation was repeated four times for a total of 96 experimental trials. The one bouncing cycle animations and the three bouncing cycles animations were presented in different blocks, the order of which was counterbalanced across participants. Within each block, the animations were presented in a random order. Before starting each block, participants were presented with all the 12 animations in a random order, to familiarize them with the task.

## Results and discussion

Response probabilities are shown in Fig 6 (one bouncing cycle) and Fig 7 (three bouncing cycles), separately for each prototypical category. In these figures, the blue cells represent animations that elicited clear physical bounce impressions (i.e., the probability of physical bounce responses was larger than .50), the red cells represent animations that elicited clear animated

## One bouncing cycle

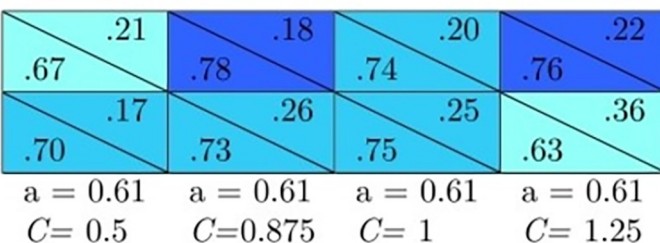

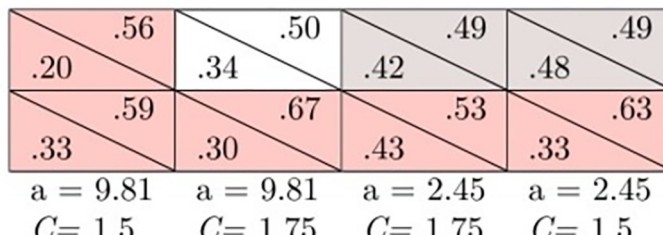

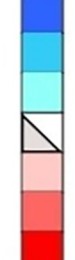

.75 < p("Physical Bounce")

.67 < p("Physical Bounce") ≤ .75

.50 < p("Physical Bounce") ≤ .67

no clear preference

.50 < p("Animated Motion") ≤ .67

.67 < p("Animated Motion") ≤ .75

.75 < p("Animated Motion")

**Fig 6.** Observed probability of physical bounce responses (lower left corner of each cell) and animated motion responses (upper right corner of each cell) for the one bouncing cycle animations of Experiment 1 (first rows) and Experiment 3 (second rows).

motion impressions (i.e., the probability of animated responses was larger than .50), and the white cells represent animations that were associated with high rates of *other* impressions.

Individual means were computed for each participant in Experiment 1 and Experiment 3 by averaging the probability of each response category (physical bounce, animated jump,

*other*) across the 24 animations that were used in both experiments. Three separate two-way mixed ANOVAs were then conducted, one for each response category, with Experiment (Experiment 1 or Experiment 3) as a between-subject factor and the number of bouncing cycles (one or three) as a within-subject factor. The ANOVA results are presented in Table 6.

The absence of a visible opaque surface resulted in a slight reduction in the mean probability of physical bounce responses (Experiment 1: $M = .45$, 95% CI [.27, .62]; Experiment 3: $M = .40$, 95% CI [.22, .57]), accompanied by a slight increase in the mean probability of animated jump responses (Experiment 1: $M = .32$, 95% CI [.15, .48]; Experiment 3: $M = .34$, 95% CI [.17, .51]) and *other* responses (Experiment 1: $M = .24$, 95% CI [.09, .39]; Experiment 3: $M = .26$, 95% CI [.17, .51]). However, despite these differences, the ANOVA results reported in Table 6 indicate that they were not statistically significant. Additionally, Figs 6 and 7 demonstrate that all animations that were considered prototypical of a physical bounce in Experiment 1 still yielded clear impressions of physical bounce in Experiment 3, and the same was true for animated jump impressions. Hence, the results support the hypothesis that the presence or absence of a visible opaque bouncing surface had a negligible effect on the resulting visual impression of bouncing or jumping.

The results reinforce the hypothesis that the visual system can automatically and postdictively infer the presence of an invisible surface from the kinematics of the moving object (see also [69]). They also provide support to the idea that the visual system can embed relatively simple motion patterns into meaningful perceptual structures characterized by definite cause-effect relationships, and that it can do so by filling in the scene with the initially missing elements (see also [4, 17, 70]).

## Experiment 4

A fundamental characteristic of visually perceived events is the presence of definite causal relationships among the objects in the scene (for a review see [1]). In this fourth experiment, we wanted to explore the characteristics of the causal impressions involved in bouncing and jumping. We hypothesize that, in the case of physical bounce impressions, the bouncing surface is seen to cause the reversal of the direction of the bouncing object. Moreover, we hypothesize that, in the case of animated jump impressions, the object's motion appears to be caused by a hidden force internal to the object itself, as if the object was a living creature propelled by self-generated motion. In the occurrence of the animated jump, the surface would therefore play a functional role for the occurrence of the event (i.e., the object *jumps* against the surface), however it would not be perceived as the cause of the direction reversal. When the motion is not perceived as a physical bounce or animated jump, the perceived cause of the direction change (if any) would not be clearly connected to the collision with the surface or to an internal force.

To test these hypotheses, we selected a range of parameters that, according to the results of the previous three experiments, gave rise to physical bounce impressions, animated motion impressions, or to no definite impression. Participants were tasked with classifying each animation based on visual impressions of causality (if any).

### Methods

**Participants.** Thirty graduate or undergraduate students at the University of Padova participated in the experiment on a voluntary basis. They were aged from 19 to 54 years ($M = 23.17$ years, 95% CI [20.67, 25.67]), 22 were women and 8 were men. All participants were naïve to the purpose of the experiment, and gave written informed consent according to the Declaration of Helsinki prior to their inclusion in the experiment. They all reported

## Three bouncing cycles

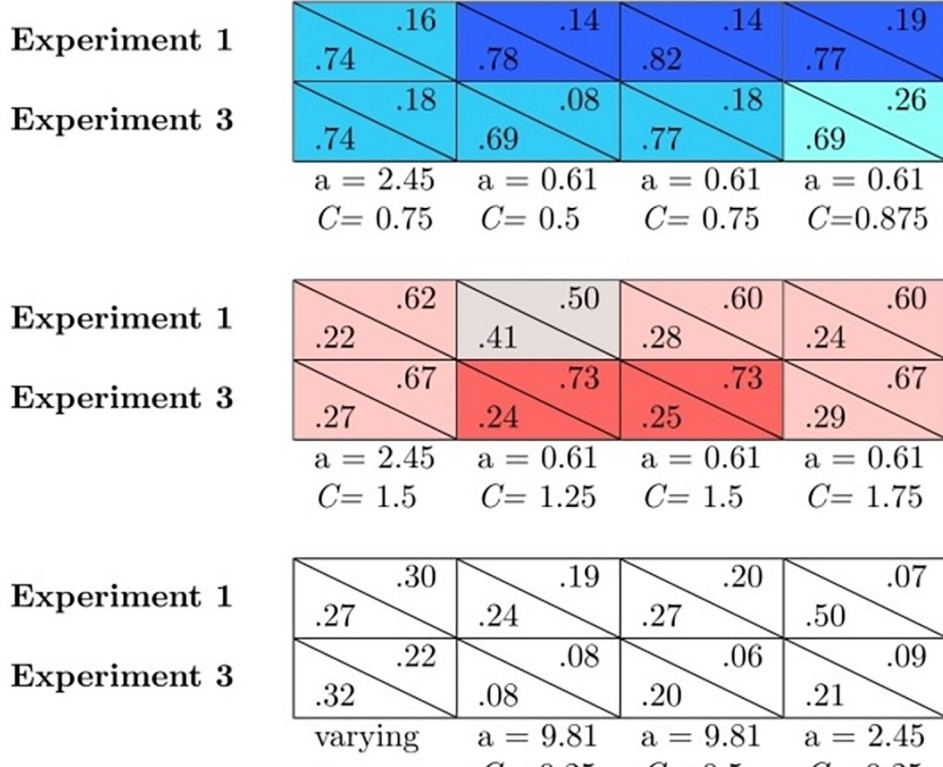

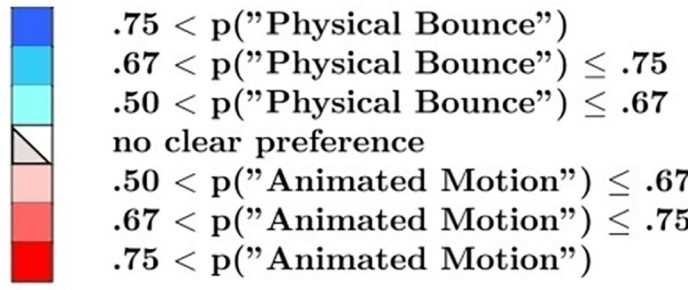

**Fig 7.** Observed probability of physical bounce responses (lower left corner of each cell) and animated motion responses (upper right corner of each cell) for the three bouncing cycles animations of Experiment 1 (first rows) and Experiment 3 (second rows).

normal or corrected-to-normal visual acuity. None of them had participated in the previous experiments. At the end of the experiment they were rewarded with 10 €.

**Stimuli and apparatus.** Everything was identical to Experiment 1, with the following exceptions. Eight different combinations of parameters $C$, $a$, and delay were selected, so as to

**Table 6. ANOVA results (Experiments 1 and 3) for the effects of experiment and number of bouncing cycles on the probability of physical bounce, animated jump, and *other* responses.**

| Physical bounce | | | | | |
|---|---|---|---|---|---|
| *Predictors* | $df_n$ | $df_d$ | *F* | *p* | $\eta_G^2$ |
| Experiment | 1 | 58 | 2.75 | .103 | .031 |
| N_bounces | 1 | 58 | 0.37 | .544 | .002 |
| Experiment × N_bounces | 1 | 58 | 1.07 | .305 | .006 |
| **Animated jump** | | | | | |
| *Predictors* | $df_n$ | $df_d$ | *F* | *p* | $\eta_G^2$ |
| Experiment | 1 | 58 | 1.14 | .290 | .014 |
| N_bounces | 1 | 58 | 1.78 | .187 | .009 |
| Experiment × N_bounces | 1 | 58 | 0.15 | .697 | < .001 |
| **Other** | | | | | |
| *Predictors* | $df_n$ | $df_d$ | *F* | *p* | $\eta_G^2$ |
| Experiment | 1 | 58 | 0.46 | .501 | .006 |
| N_bounces | 1 | 58 | 0.34 | .563 | .001 |
| Experiment × N_bounces | 1 | 58 | 2.99 | .089 | .010 |

**Note**: $df_n$ = degrees of freedom at the numerator of the F statistic; $df_d$ = degrees of freedom at the denominator of the *F* statistic; *F* = value of the *F* statistic; *p* = *p*-value (boldface type indicates a statistically significant effect); $\eta_G^2$ = generalized eta squared.

generate a broad range of impressions from physical bounce, to animated jump, to *other*. The eight combinations of parameters are reported in Fig 8. For consistency, all the animations showed uniform acceleration as motion pattern. These same set of eight combinations of parameters were used to generate one bouncing cycle animations and three bouncing cycle animations. Moreover, for the sake of generality, each animation could be presented with the visible opaque bouncing surface, as in Experiments 1 and 2, or with no visible opaque bouncing surface, as in Experiment 3. Therefore, a total of 32 animations [i.e., 2 number of bouncing cycles (1 or 3) × 2 opaque bouncing surface (present or absent) × 8 combinations of parameters] were used as stimuli in Experiment 4.

**Procedure.** Participants were randomly divided into two different groups (N = 15 per group). One group was presented with the animations in which the visible opaque bouncing surface was included, whereas the other group was presented with the animations showing no visible opaque surface. Participants were always tested individually.

The procedure was the same as in Experiment 1, with the following exceptions. Participants who were shown the animations with the visible opaque surface were given the following instructions: "a) external physical cause (key 1): the motion direction change of the disk seems to be caused by the contact with the surface against which the object appears to bounce; b) internal psychological cause (key 2): the motion direction change of the disk appears to be caused by a force internal to the objects itself, as if it were an active object with its own internal energy; c) *other* (key 3): the motion direction change of the disk does not appear to be caused by an external cause (surface) or by an internal cause (internal energy)." The same instructions were also used for the participants who were presented with the animations with no visible opaque bouncing surface, except that "transparent surface" was used instead of "surface".

Each animation was repeated five times, for a total of 80 experimental trials for each group of participants. In both groups, the one bouncing cycle animations and the three bouncing cycles animations were presented in different blocks, the order of which was counterbalanced across participants. Within each block, the animations were presented in a random order.

## One bouncing cycle

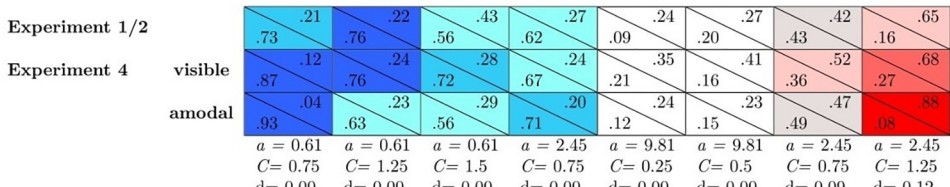

## Three bouncing cycles

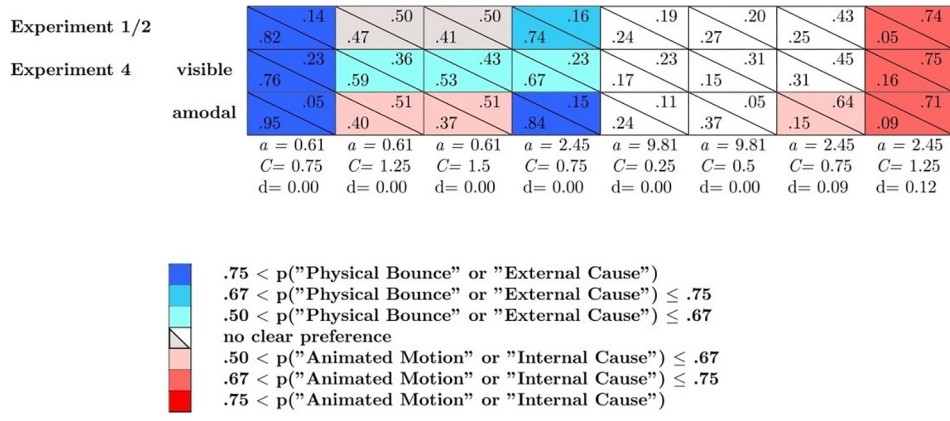

**Fig 8.** Observed probability of physical bounce or external cause responses (lower left corner of each cell) and animated motion or internal cause responses (upper right corner of each cell) for the animations of Experiment 1/2 (first rows) and Experiment 4 (second/third rows).

Before starting each block, participants were shown the corresponding six animations twice in a random order, to familiarize them with the task.

## Results and discussion

In the two tables in Fig 8, the first row shows the probabilities of physical bounce (lower left corner) and animated motion responses (upper right corner) that were observed in Experiments 1 and 2. Blue and red colors indicate clear impressions of physical bounce and animated motion, respectively. In both tables, the second and the third row represent the results from Experiment 4, showing the probability of external physical cause responses (lower left corner) and internal psychological cause responses (upper right corner). The second row refers to the animations showing the visible opaque surface, whereas the third row refers to the animations with no visible opaque surface. In both rows, blue and red colors stand for clear impressions of external physical cause and for clear impressions of internal psychological cause, respectively.

In line with our hypotheses, Fig 8 shows that the animations in Experiment 1/2 that had given rise to clear physical bounce impressions, gave rise to clear impressions of external physical causality. Symmetrically, the animations in Experiment 1/2 that had given rise to clear animated motion impressions gave rise to clear impressions of internal psychological causality.

As for the statistical analyses, we first explored if the presence or absence of the opaque surface had any effect on the reported impressions of causality. In Experiment 4, we calculated the

mean probability of each response category (external physical cause, internal psychological cause, *other*) for each participant by averaging their responses to the 16 animations used in both conditions of the experiment (i.e., opaque surface and amodal surface). We then conducted three separate two-way mixed ANOVAs (one for each response category) with Surface (opaque surface or amodal surface) as a between-subject factor and the number of bouncing cycles (one or three) as a within-subject factor. The results, presented in Supplementary Table S8 on OSF, indicate no statistically significant main or interaction effects for the Surface factor.

The results of this preliminary analysis indicate that, overall, the presence or absence of the opaque surface had little effect on physical and psychological causality impressions. Therefore, for the following analyses, the response probabilities were averaged across the presence/absence of the visible opaque surface (i.e., from 32 to 16 different animations).

For each participant in Experiment 4, the probability of each response category (external physical cause, internal psychological cause, *other*) was calculated based on the 16 animations used, without distinguishing between opaque and amodal surfaces. Similarly, the probability of each response category (physical bounce, animated motion, *other*) was calculated for each participant in Experiment 1 based on the 12 animations used in both Experiment 1 and Experiment 4, and for each participant in Experiment 2 based on the four animations used in both Experiment 2 and Experiment 4. To analyze the data, three two-way mixed ANOVAs were performed, one for each response category (physical bounce/external physical cause, animated motion/internal psychological cause, *other*), with Experiment (Experiments 1/2 or Experiment 4) as a between-subject factor and the number of bouncing cycles (one or three) as a within-subject factor. The ANOVA results are presented in Table 7.

For the set of animations here considered, the mean probability of external physical cause responses in Experiment 4 tended to be larger than the mean probability of physical bounce responses in Experiments 1/2 (Experiments 1/2: $M$ = .35, 95% CI [.23, .47]; Experiment 4: $M$ = .45, 95% CI [.23, .63]). Moreover, the mean probability of internal psychological cause responses in Experiment 4 tended to be slightly lower than the mean probability of animated

**Table 7. ANOVA results (Experiments 1/2 and 3) for the effects of experiment and number of bouncing cycles on the probability of physical bounce/external physical causality, animated motion/internal psychological causality, and *other* responses.**

| Physical bounce | | | | | |
|---|---|---|---|---|---|
| Predictors | $df_n$ | $df_d$ | $F$ | $p$ | $\eta_G^2$ |
| Experiment | 1 | 88 | 6.64 | **.012** | .056 |
| N_bounces | 1 | 88 | 16.20 | **< .001** | .039 |
| Experiment × N_bounces | 1 | 88 | 0.54 | .466 | .001 |
| **Animated jump** | | | | | |
| Predictors | $df_n$ | $df_d$ | $F$ | $p$ | $\eta_G^2$ |
| Experiment | 1 | 88 | 2.79 | .099 | .025 |
| N_bounces | 1 | 88 | 2.11 | .149 | .005 |
| Experiment × N_bounces | 1 | 88 | 0.19 | .662 | < .001 |
| **Other** | | | | | |
| Predictors | $df_n$ | $df_d$ | $F$ | $p$ | $\eta_G^2$ |
| Experiment | 1 | 88 | 0.48 | .492 | .004 |
| N_bounces | 1 | 88 | 6.61 | **.012** | .016 |
| Experiment × N_bounces | 1 | 88 | 0.07 | .786 | < .001 |

**Note**: $df_n$ = degrees of freedom at the numerator of the F statistic; $df_d$ = degrees of freedom at the denominator of the $F$ statistic; $F$ = value of the $F$ statistic; $p$ = p-value (boldface type indicates a statistically significant effect); $\eta_G^2$ = generalized eta squared.

motion responses in Experiments 1/2 (Experiments 1/2: $M = .42$, 95% CI [.30, .55]; Experiment 4: $M = .35$, 95% CI [.18, .52]). A negligible difference was instead observed for the mean probability of *other* responses (Experiments 1/2: $M = .23$, 95% CI [.12, .33]; Experiment 4: $M = .20$, 95% CI [.06, .35]). The ANOVA results presented in Table 7 indicate that the only statistically significant difference observed was between physical bounce responses and external physical causality responses.

Regarding the observed difference, it is important to note that even in cases of animated motion, there is typically some apparent physical interaction between the animated object and the bouncing surface. For instance, a living entity jumping on a surface would physically interact with the surface. Apparently, the participants in Experiment 4 had a slight tendency to give more importance to the external physical cause component than to the internal psychological one. However, despite these differences, Fig 8 shows that all animations that were considered prototypical of a physical bounce in Experiments 1/2 yielded strong impression of external physical causality in Experiment 4, and all animations that were considered prototypical of an animated motion in Experiments 1/2 yielded strong impression of internal psychological causality in Experiment 4. We also note that, across the 16 animations, the correlation between the probability of physical bounce responses and the probability of external physical causality responses was strong and positive ($r = .97$, 95% CI [.90, .99]), as well as that between the probability of animated motion responses and the probability of internal psychological causality responses ($r = .91$, 95% CI [.76, .97]). These results confirm that physical bounce impressions were associated with the impression that the contact with the bouncing surface caused the direction reversal of the disk, and that animated motion impressions were associated with impressions of internal psychological causality.

## General discussion

A typical bounce event is generally described as a perceptual structure characterized by the multiple repetition of a basic unit, i.e. a bouncing cycle. The latter is composed by a falling phase, a collision against a bouncing surface, and a climbing-back phase. The first aim of the present work was to compare the laws underlying the visual perception of physical motion and animated motion in bouncing-like scenarios with the physical laws of bouncing. In Experiments 1 and 2, several low-level features of the scene were manipulated, namely the motion pattern (i.e., presence or absence of uniform acceleration/deceleration), the simulated value of gravitational acceleration, the coefficient of restitution $C$, the number of bouncing cycles, and the delay at the moment of collision. Experiment 3 also tested the possible effects of removing the visible opaque bouncing surface on the perception of physical bounce and animated motion. Lastly, Experiment 4 explored the relationship between physical bounce and animated motion impressions on one hand, and the corresponding impressions of causality on the other. The theoretical implications of the results will be discussed in the remainder of the article.

### The modulation effect of the number of bouncing cycles

The results of Experiment 1 showed that, in the case of three bouncing cycles, the reported impression of physical bounce or animated motion was strongly affected by the physical plausibility of $C$. Indeed, physical bounce impressions tended to emerge when $C < 1$, and animated motion impressions tended to emerge when $C > 1$. On the contrary, participants' responses were relatively unaffected by the physical plausibility of the motion pattern (i.e., uniform velocity or uniform acceleration). Interestingly, the opposite pattern of results emerged in the case of one bouncing cycle, as the presence of uniform acceleration/deceleration was a relatively

strong cue to physical bounce whereas the physical plausibility of $C$ was relatively unimportant.

To provide a tentative explanation for the modulation effect of the number of bouncing cycles, we start by noting that previous studies indicate that the perception of the elasticity of bouncing objects is driven by the ratio of the peak heights [35–37]. Peak heights tend to decrease if $C < 1$, to remain constant if $C = 1$, and to increase if $C > 1$ (see Eq 6). Three bouncing cycles animations were characterized by four different peak heights (i.e., the initial height, and the peak heights after the first, second, and third bounce), implying a progressive decrease of the peak heights when $C < 1$ and a progressive increase when $C > 1$. Therefore, the visual information about the plausibility of $C$ was redundant. Instead, there were just two peak heights in one bouncing cycle animations, namely the initial height and the height after the first bounce. In other words, visual information about the physical plausibility of $C$ was clearly more abundant (i.e., more redundant) in the case of three bouncing cycles animations than in the case of one bouncing cycle animations. The same reasoning would hold true if the crucial variable for the perception of the plausibility of physical bounces were the speeds ratio or the periods ratio (see Eq 6). We speculate that, for animations depicting only one bouncing cycle, the visual information regarding parameter $C$ might not have been presented redundantly enough to function as a reliable indicator of its physical plausibility.

Interestingly, motion pattern (i.e., presence or absence of uniform acceleration/deceleration) exerted a relatively strong influence on visual impressions of physical bounce and animated motion in the case of one bouncing cycle, that is, precisely when the optical information about $C$ (i.e., the ratio of peak heights) was less redundant, and thus presumably less reliable. In light of these results, it is possible to hypothesize that motion pattern is a secondary visual cue to the physical plausibility of the bounce, that comes in the foreground only when the visual information about the primary cue (i.e., the plausibility of $C$) is not sufficiently reliable, as in the case of one bouncing cycle animations.

## Physical and perceptual constraints

According to the noisy Newton model, the perception of events would be driven by internalized physical constraints [26–30]. However, the results of Experiment 1 highlight some important differences between physical constraints and the representations of these constraints. From a physical viewpoint, all the relevant physical laws are independent of the number of bouncing cycles. However, at a perceptual level, one bouncing cycle and three bouncing cycle animations are clearly not the same. Although physical constraints like energy conservation and gravitational acceleration are ubiquitous, a realistic representation of energy conservation emerged for three bouncing cycles animations but not for one bouncing cycle animations, whereas the opposite tended to be true for the representation of gravitational acceleration.

In line with the hypothesis upheld by Kominsky et al. [31], we suggest that perceptual constraints impose limits on the number and the precision of the constraints that can be internalized in a given physical scenario. Accurate representations of the energy conservation constraint are more likely to emerge in scenarios showing three or more bouncing cycles because, as highlighted in the previous section, the visual information about this constraint is redundant in those scenarios. Realistic representations of energy conservation are less likely to emerge in one bouncing cycle scenarios, mainly because these scenarios provide little visual information about possible energy conservation or violation. This suggests that the constraints internalization is context-specific, that is, it depends on the contextual perceptual features of the scenario, and the constraints that are internalized in a given scenario may not extend to scenarios that are similar to it from a physical viewpoint.

Some of the results from Experiments 1 and 2 appear to run against the hypothesis that the perception of visual events is driven by internalized physical constraints. First, physical bounce impressions were more likely to emerge for relatively low acceleration values of $g/4$ and $g/16$ than for the physically correct value $g$. Surprisingly, none of the animations showing acceleration/velocity values consistent with terrestrial gravitational acceleration (i.e., $g = 9.81$ m/s$^2$) gave rise to a clear physical bounce impression. This result appears to be inconsistent with the hypothesis of an internalized representation of terrestrial gravitational acceleration [71]. It is worth noting that the physical acceleration of a falling object can produce different retinal acceleration values depending on the object's distance from the observer. As per the inverse relationship between retinal acceleration and distance [72], a small object located nearby and falling with a physically unrealistic low acceleration value could produce the same retinal acceleration as a large object located far away falling with a physically plausible acceleration of $g$. Given that our stimuli lacked absolute cues for size and distance, it is possible that the observers could have perceived accelerations of $g/4$ and $g/16$ as instances of large distant objects falling with a physically accurate acceleration of $g$, instead of a small object close to the observer's viewpoint falling with an implausible physical acceleration. Nevertheless, recent studies have shown that terrestrial gravitational acceleration is largely underestimated even when observers are presented with real-scale objects and environments [60, 61], which appears to suggest that observers have a spontaneous preference for acceleration/deceleration values smaller than $g$. Preference for small acceleration/deceleration values might be related to the fact that a slightly slowed playback of an event may increase its visibility, offering the possibility to better grasp the event in its details. It is also worth adding that, in the light of the results of the present study, it cannot be ruled out that the impression of natural or unnatural motion may reflect a preference for velocity rather than a preference for acceleration. This view is consistent with data showing that observers do not exhibit a clear preference for uniformly accelerated motion over uniform velocity motion when judging the plausibility of a vertical fall [58].

As an additional note of caution for the interpretation of the results concerning the perceptual plausibility of stimuli with low acceleration values, air resistance may have affected the outcomes. Eqs 1–2 and 4–5 assume no impact from air resistance, but on Earth, objects tend to fall slightly slower than the nominal gravitational acceleration due to this factor. This effect is more pronounced for relatively light objects and large falling heights, as opposed to heavier objects and shorter distances (for a detailed discussion, see [61]). Since our stimuli did not include information about mass and distance, it is possible that observers assumed a very light object falling from a significant height. In this case, the falling speed would be lower than $g$ from a physical standpoint, taking air resistance into account. However, it is also important to note that previous research has shown that the effects of air resistance are not dramatically large even for light objects, as demonstrated in [58]. For instance, on Earth, a polystyrene sphere weighing 5 grams falling from a height of 2.20 meters would experience a mean acceleration of 8.567 m/s$^2$, or $g/1.15$.

Another result which is not consistent with the hypothesis of internalized physical constraints is that, despite their being physically implausible, relatively long delays of 60 ms at the moment of the collision between the disk and the bouncing surface did not disrupt physical bounce impressions. Furthermore, delays of 30 ms even improved them slightly. These results mirror Michotte's [2] findings about the effects of time delays on the launching effect, and suggest that the perceived temporal contiguity of the falling and the climbing back phase is a key factor for the visual perception of physical bounce, above and beyond physical realism.

Overall, the results appear to indicate that, besides being driven by internalized physical constraints like energy conservation and gravitational acceleration, visual perception of bouncing is also driven by genuine perceptual constraints like preference for acceleration values

smaller than *g* and perceived temporal contiguity. This further reinforces the idea that the visual perception of events is driven by the interplay between physical and perceptual constraints (see also [31]).

## Phenomenal causality and impetus transmission

Visually perceived events are typically associated with impressions of phenomenal causality, that is, with the impression that the motion of one or more objects is caused by the interaction with another object [1]. The results of Experiment 4 show that bouncing and jumping are associated with definite impressions of causality. In the case of physical bounce, the surface (opaque or amodal) was perceived as the immediate cause of the reversal of the trajectory of the bouncing object. When animated motion was instead perceived, the surface played the functional role of an entity that allowed to the object to jump away, but with *its own force*.

Bouncing adds to the set of events that generate visual impression of causality. White [73, 74] proposed a general theoretical framework for the perception of causality, according to which causal events would be intrinsically related to a perceived asymmetry between an active object (i.e., an *agent*) and a passive object (i.e., a *patient*). In this framework, the agent is the object that moves first, whereas the patient is the object that starts moving after the contact with the active object. Causality would be perceived when the properties of the agent are seen to be transmitted to the patient (see also [11]). In a similar vein, Hubbard [23] suggested that visual perception of causality is driven by an impetus transmission heuristic, by which an impetus is transmitted from an agent to a patient (see also [24, 44]). One fundamental characteristic of the impetus transmission heuristic is that, after the impetus is imparted by the agent to the patient, it gradually dissipates, leading to a gradual slowdown of the patient's motion. Another fundamental characteristic of this heuristic is the conservation of impetus, that is, if an impetus is transmitted from *A* to *B*, then the impetus of *B* cannot be greater than the impetus of *A*. According to Hubbard [23], a causal relationship would be perceived between two objects *A* and *B* when the kinematic pattern of these objects is consistent with an impetus transmission heuristic.

In the case of bouncing, it would be intuitively obvious to identify the disk as the agent and the bouncing surface as the patient, because the disk is the only object that moves in the scene. However, at odds with the predictions from White's [73, 74] theoretical framework as well as with the predictions from the impetus transmission heuristic [23], it is not the agent that is perceived to cause the motion of the patient, but rather it is the stationary bouncing surface (i.e., the patient) that appears to exert a causal effect on the disk (i.e., the agent), by making its direction change upon contact. In this respect, causal impressions involved in physical bounce are similar to those involved in the braking effect [8] and in the shattering effect [13], in which a passive stationary object appears to exert a causal effect on the motion of an initially moving object. This means that, under appropriate conditions, a stationary object can be seen to cause the behavior of an initially moving object (see also [64, 75]). It appears to be possible to reconcile this empirical evidence with the impetus transmission heuristic by assuming that there are circumstances in which impetus can be transmitted not only forwards from an agent to a patient, but also backwards from the patient to the agent (i.e., bidirectional impetus transmission). A bidirectional impetus transmission heuristic would be used when the patient is perceived to be much more massive than the agent [76]. Cues to the massiveness of the patient can be provided by its size, the apparent material of which it is made or, in line with the KSD principle [77], even by the kinematic behavior of the agent (e.g., an object that bounces back or shatters after the contact with another object provides a visual cue to the massiveness of the latter).

The results of our experiments also indicate that, besides being driven by general heuristics like impetus or property transmission, participants' responses to a given scenario also depend on the perceptual constraints that operate in that scenario. For instance, taken alone, the impetus transmission heuristic cannot explain why animations characterized by a simulated gravitational attraction of *g* were never perceived as natural physical bounces. Moreover, because impetus conservation is one distinctive feature of the heuristic [23], the heuristic itself cannot fully explain why one bouncing cycle animations showing values of *C* well larger than 1 still gave rise to impressions of physical bounce.

## Animated jumps

As several authors argued, the ability of the visual system in discriminating between physical movements and animated movements is an adaptive behavior with important implications for survival. For this reason, it has to be largely automatic and deeply rooted in early visual processes [5]. In our experiments, the most compelling animated jumping-like motions tended to occur when the stimuli showed a clear violation of energy conservation, which provides support for the hypothesis that the inconsistency with the physical laws is a crucial factor in determining animacy impressions (e.g., [51, 52]). It is however worth remarking that, albeit necessary, the violation of Newtonian laws was not sufficient for the occurrence of jumping impressions. Indeed, in the case of one bouncing cycle, even animations showing clear violations of energy conservation led to visual impressions of physical bounce. As previously discussed, this is probably related to the fact that, in the case of one bouncing cycle, observers were relatively insensitive to energy conservation violations.

Interestingly, our data also show that there was no clear-cut dichotomy between physical bounce and animated motion impressions, but rather a continuous transition (see Figs 2–4). As the probability of perceiving physical bounce decreased, the probability of perceiving an animated motion increased. Therefore, there was a range of temporal and kinematic parameters that could give rise to both the one and the other impression. A possible reason could be that a jump is not totally dissimilar to a physical bounce. Rather, the former can be described as a bio-mechanical event, in which an internal force, a surplus of impetus, appears to add to the impetus that gradually dissipates. The transition from physical bounce to animated jump would reflect a gradual increase in the perceived amount of this additional impetus.

This gradual transition from bouncing to jumping is reminiscent of the gradual transition from launching to triggering [2], which is characterized by the fact that reports of launching (i.e., physical motion) gradually give way to reports of triggering (i.e., self-generated motion) as the relative speed of *B* increases with respect to the speed of *A*. Therefore, beyond Hafri and Firestone's [4] claim that the perception of physical and social events is categorical, our data, as well as those reported by Michotte [2], appear to indicate that physical and animated events can be better understood not as opposite categories but as the two poles of a continuum, by admitting the possibility of intermediate categories which could be perceptually meaningful. As can be seen with some visual illusions [78], the ambiguity between physical bounce and animated bounce could itself be relevant from a biological perspective. That is, to deceive predators or prey, a living organism could "mask" the biological character of its movement, giving a more physical-like character to the motion itself. At the same time, organisms would get an adaptive advantage to unmask perceptual ambiguity, and to be therefore attracted by it.

In other words, the existence of events characterized by a blend of physical and animated motion has to be acknowledged. After all, many events of daily life, such as the rambling movement of a leaf blown by the wind, look ambiguous and not immediately identifiable as animate or inanimate. This sort of event plausibly constitutes a hybrid category, whose visible

ambiguity can arouse surprise, curiosity and motivate greater exploration of the event, as suggested by the surreal episode of the bouncing ball described in the incipit of this work.

## Author Contributions

**Conceptualization:** Michele Vicovaro, Giulia Parovel.

**Data curation:** Michele Vicovaro, Loris Brunello.

**Formal analysis:** Michele Vicovaro, Loris Brunello.

**Funding acquisition:** Giulia Parovel.

**Investigation:** Michele Vicovaro, Loris Brunello, Giulia Parovel.

**Methodology:** Michele Vicovaro, Giulia Parovel.

**Software:** Michele Vicovaro.

**Visualization:** Michele Vicovaro, Loris Brunello.

**Writing – original draft:** Michele Vicovaro, Giulia Parovel.

**Writing – review & editing:** Michele Vicovaro, Giulia Parovel.

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
