## [Decision Letter · Decision Letter 0]

16 Jan 2023

PONE-D-22-33138The psychophysics of bouncing: perceptual constraints, physical constraints, animacy, and phenomenal causalityPLOS ONE

Dear Dr. Parovel,

Thank you for submitting your manuscript to PLOS ONE. After careful consideration, we feel that it has merit but does not fully meet PLOS ONE’s publication criteria as it currently stands. Therefore, we invite you to submit a revised version of the manuscript that addresses the points raised during the review process. As you will read, reviewer 1 suggested a minor revision, and made a few conceptual or methodological comments. Please answer these comments to solve the issues which were raised. The reviewer 2 however suggested a major revision, mainly focused on the statistical analysis. The reviewer made an extensive description of his concerns and re-analyzed part of your data to strengthen his claim. He eventually made suggestions on what analysis should be performed given all the issues raised. It seems to me that given the analysis suggestion, all the issues can be answered, with a strong statistical re-analysis.

We look forward to receiving your revised manuscript.

Kind regards,

Robin Baurès, Ph.D.

Academic Editor

PLOS ONE

Journal Requirements:

2. Please change "female” or "male" to "woman” or "man" as appropriate, when used as a noun (see for instance https://apastyle.apa.org/style-grammar-guidelines/bias-free-language/gender).

Reviewers' comments:

Reviewer's Responses to Questions

**Comments to the Author**

1. Is the manuscript technically sound, and do the data support the conclusions?

Reviewer #1: Yes

Reviewer #2: Yes

2. Has the statistical analysis been performed appropriately and rigorously? 

Reviewer #1: Yes

Reviewer #2: No

3. Have the authors made all data underlying the findings in their manuscript fully available?

Reviewer #1: Yes

Reviewer #2: Yes

4. Is the manuscript presented in an intelligible fashion and written in standard English?

Reviewer #1: Yes

Reviewer #2: Yes

5. Review Comments to the Author

Reviewer #1: Review of “The psychophysics of bouncing: perceptual constraints, physical constraints, animacy, and phenomenal causality”

The authors report four experiments that investigate the perceptual impressions of simulated bouncing events, which were manipulated to be consistent or inconsistent with Newton’s laws. A circular shape moved downwards on a computer screen and then back upwards for different distances and at different acceleration/deceleration patterns. It turned out that impossibly high restitution coefficients of the animated circle produced impressions of animacy, whereas coefficients in the possible range produced judgments of a ball bouncing. The latter responses were collected in terms of 3AFC counts. A particularly interesting side finding is that it matters little whether or not the floor upon which the ball bounces is visually present or absent.

The intuitive physics questions at the outset of the study are well presented and well motivated. The manuscript is well written, the methods are sound and the interpretation of the data is fair. The manuscript amounts to a nice addition to the field. I recommend that the substantial and well crafted paper be published. However, there are a few points that should be addressed.

Major points:

The duration of the simulated event may be critical. Given the paucity of the stimulus display, there is no independent information about size and distance of the moving object. For real-world objects, the acceleration could in principle be used to infer this information, however, in the context of the present doctored events, this is no longer possible. This is important because Newton’s laws neglect air resistance. They latter is ubiquitous on our planet, and massively affects very light objects, and can often be neglected for heavy objects. Given the unadorned stimulus, air effects (be they assumed, anticipated, or perceived) may well have played an important role in the current experiment. The restitution above 1 is of course impossible in all cases, but the average speed may or may not be considered impossible depending on air resistance. Thus, it is important to put the effects, in particular that of constant velocity, into perspective. The potential role of air should be discussed. This leads up to a second point:

The choice of the 3AFC paradigm should be better motivated. One could get more quantitative information out of naturalness ratings. The rationale for choosing 3AFC should be better laid out in the introduction and discussed at the end. Would a 2AFC paradigm been possible? Picking the “better” event when showing two bounces side-by-side could also have been an option. Note, I am not suggesting to redo things with a different paradigm, but the pros and cons of the one used here should become a bit more transparent.

Minor points:

Methods: In Exp. 1, it should be stated clearly if there was a time delay between the moment the front edge of the circle touched the floor and the begin of its rebound.

In Exp. 2 the authors used uniform acceleration at ¼ g. Why was this clearly unnatural gravitational constant used? This slow-motion of a dropping event may not only fail to be representative of real bounces, it may also allow viewers to use visual cues that may not be accessible at normal g.

As a note, maybe to the journal more than to the author, I found it quite annoying to have to scroll all the way down to see the figures. Why not integrate them into the text? In this day and age there are not disadvantages, only usability advantages.

Figure 2, 3, and 4, also 8 and 9, are messy and they do not summarize the results In any intuitive visual way. I would relegate them into an appendix.

Lines 471ff: The time it takes a tennis ball to deform and rebound (5 ms) may not be a convincing argument that the delay times of 30 ms and above are impossible. Take a large ball that is not inflated to its maximum. I bet that rebound times around 30 ms are possible. This aside, the visual system integrates sudden onset movement over even larger time windows (see e. g. Runeson, S. 1974, Constant velocity—not perceived as such. Psychological Research, 37(1), 3-23). Would the authors argue that this is what happens as well in the case of bouncing events?

The sentence starting on line 871 “We speculate …” is awful and flawed. “Cue of the physical plausibility” makes no sense.

Line 887: The sentence starting with “From a physical viewpoint bouncing is a unitary event …” is misleading. A falling object is a lot simpler to describe than a bouncing object. I guess what is meant here is that a truncated event, i. e. one bounce only, may look odd although it simpler than the complete event.

Line 909: The statement that acceleration changes as a function of distance to the event is incorrect. Just because the visual angle (and retinal size) of an object changes with distance, the acceleration does not. A Martian g would be very strange for a terrestrial environment, no matter how far away the observer is from the falling object.

Signed review

Heiko Hecht

Reviewer #2: This study reports 4 experiments investigating which visual parameters (kinematics, presence or not of a bouncing surface) produce physical or animacy (or neither) interpretations of a bouncing motion. Although I found most of the experimental manipulations relevant and I may agree with the theoretical framing used for interpreting the findings, I think as it stands, the statistical analyses do not do the readers any favors or do the data justice in order to help draw out the essence of the data. That is why the bulk of my report will focus on these analyses, and I recommend a major revision.

See details in the attached pdf file

6. PLOS authors have the option to publish the peer review history of their article (what does this mean?). If published, this will include your full peer review and any attached files.

Reviewer #1: No

Reviewer #2: No

---

## [Author Response · Author response to Decision Letter 0]

27 Mar 2023

RESPONSES TO THE COMMENTS FROM THE EDITOR

E1: Thank you for submitting your manuscript to PLOS ONE. After careful consideration, we feel that it has merit but does not fully meet PLOS ONE’s publication criteria as it currently stands. Therefore, we invite you to submit a revised version of the manuscript that addresses the points raised during the review process.

 AU1: We would like to thank the Editor for the positive appraisal of our manuscript. The insightful comments provided by the reviewers have led to several changes in the revised version of the manuscript, which we believe have significantly improved the quality of the research presented. We hope that these revisions have met the high standards set by this valuable journal and hope that the manuscript will be deemed suitable for publication.

E2: As you will read, reviewer 1 suggested a minor revision, and made a few conceptual or methodological comments. Please answer these comments to solve the issues which were raised.

 AU2: We are grateful for the constructive feedback provided by Reviewer 1, which helped us to improve and clarify the discussion of important methodological and theoretical issues. We have addressed each of the reviewer's comments and our responses can be found below.

E3: The reviewer 2 however suggested a major revision, mainly focused on the statistical analysis. The reviewer made an extensive description of his concerns and re-analyzed part of your data to strengthen his claim. He eventually made suggestions on what analysis should be performed given all the issues raised. It seems to me that given the analysis suggestion, all the issues can be answered, with a strong statistical re-analysis.

 AU3: We extend our gratitude to Reviewer 2 for providing a thorough and detailed review of our statistical approach. As per his recommendations, we have re-analyzed the data from all four experiments using repeated measures ANOVAs on response proportions (with Huynh-Feldt correction) instead of GLMM. Furthermore, we have updated the results section of each experiment in accordance with the changes made (please refer to our response to Reviewer's comment 3 for a comprehensive outline of these updates). We would like to highlight that, thanks to the reviewer's insightful feedback and suggestions, we believe that the revised manuscript better adheres to a standard approach to data analysis and representation, and offers greater clarity. Importantly, we would also like to highlight that none of the main conclusions drawn from our study have changed following the re-analysis, which reinforces the robustness of our results and conclusions.

We would like to express our sincere appreciation to you and the reviewers for the valuable and rigorous feedback provided, which has significantly improved the quality of our work.

RESPONSES TO THE COMMENTS FROM REVIEWER #1

R1: The intuitive physics questions at the outset of the study are well presented and well motivated. The manuscript is well written, the methods are sound and the interpretation of the data is fair. The manuscript amounts to a nice addition to the field. I recommend that the substantial and well crafted paper be published. However, there are a few points that should be addressed.

 AU1: We are grateful to the reviewer for his positive appraisal and thoughtful suggestions. We have made every effort to address all of the concerns raised by the reviewer in the revised manuscript and in our responses to his comments. We appreciate the reviewer's time and effort in providing feedback that has helped us to improve our work.

Major points

R2: The duration of the simulated event may be critical. Given the paucity of the stimulus display, there is no independent information about size and distance of the moving object. For real-world objects, the acceleration could in principle be used to infer this information, however, in the context of the present doctored events, this is no longer possible. This is important because Newton’s laws neglect air resistance. They latter is ubiquitous on our planet, and massively affects very light objects, and can often be neglected for heavy objects. Given the unadorned stimulus, air effects (be they assumed, anticipated, or perceived) may well have played an important role in the current experiment. The restitution above 1 is of course impossible in all cases, but the average speed may or may not be considered impossible depending on air resistance. Thus, it is important to put the effects, in particular that of constant velocity, into perspective. The potential role of air should be discussed. This leads up to a second point:

 AU2: We appreciate the reviewer's comment, as it allowed us to examine the potential relationship between air resistance and the findings of our experiments. We have provided a detailed discussion of this important issue in the revised manuscript (see lines 1301-1311 of the manuscript with track changes). While we agree with the reviewer that it is important to carefully describe and consider the role of air resistance, we also note that previous studies have indicated that its effects are generally not particularly large, even in the case of light objects with high falling distances.

R3: The choice of the 3AFC paradigm should be better motivated. One could get more quantitative information out of naturalness ratings. The rationale for choosing 3AFC should be better laid out in the introduction and discussed at the end. Would a 2AFC paradigm been possible? Picking the “better” event when showing two bounces side-by-side could also have been an option. Note, I am not suggesting to redo things with a different paradigm, but the pros and cons of the one used here should become a bit more transparent.

 AU3: We would like to express our appreciation to the reviewer for his comment, as we share his belief that discussing our methodological choices can enhance the transparency of our work. We have provided a detailed discussion of the reasons behind our selection of a 3AFC task and the comparison with alternative techniques in the revised manuscript (see lines 339-350 of the manuscript with track changes). In short, we chose to use a third response alternative ("other") to prevent ambiguous stimuli that could not be clearly classified as either physical bounce or animated motion from increasing the diversity of visual impressions within the two primary response categories, which could have led to greater uncertainty in their interpretation. Additionally, we opted for the 3AFC task over rating scales or paired comparisons due to its brevity and simplicity.

Minor points

R4: Methods: In Exp. 1, it should be stated clearly if there was a time delay between the moment the front edge of the circle touched the floor and the begin of its rebound.

 AU4: Thank you for pointing this out. On lines 298-300 of the manuscript with track changes we have highlighted that there was no time delay between the moment the front edge of the circle touched the bouncing surface and the start of the climbing-back phase.

R5. In Exp. 2 the authors used uniform acceleration at ¼ g. Why was this clearly unnatural gravitational constant used? This slow-motion of a dropping event may not only fail to be representative of real bounces, it may also allow viewers to use visual cues that may not be accessible at normal g.

 AU5: As it now underlined on lines 656-658 of the manuscript with track changes, this was done because the results from Experiment 1 had shown that g/4 was a suitable acceleration value for creating clear visual impressions of physical bounce and animated motion. The same empirical results had shown that a normal g acceleration was incompatible with clear physical bounce impressions.

R6: As a note, maybe to the journal more than to the author, I found it quite annoying to have to scroll all the way down to see the figures. Why not integrate them into the text? In this day and age there are not disadvantages, only usability advantages.

 AU6: We completely agree with the Reviewer on this point, but we adhered to the journal submission guidelines. However, to facilitate the reviewing process, we have now uploaded all the figures on OSF, which can be accessed through the following link: https://osf.io/xz8t3/?view_only=741328b1d38b4f8d9d72bbfbe745ba85. This link includes not only the supplementary figures but also the "regular" figures. By doing so, we hope to make it easier for the Reviewer to view each figure on a separate page without having to scroll down each time.

R7: Figure 2, 3, and 4, also 8 and 9, are messy and they do not summarize the results In any intuitive visual way. I would relegate them into an appendix.

 AU7: In response to the Reviewer's suggestion, we have relocated Figures 2, 3, and 4 to OSF, where they are now available as supplementary figures. We have also replaced Figures 2 and 3 with line graphs, which we believe provide a more intuitive representation of the results. However, we would like to keep Figures 8 and 9 (now labelled as Fig 6 and 7) in the manuscript, as we believe that they allow for a direct comparison between the results from Experiment 3 and Experiments 1. These figures enable the reader to compare cell colors and observe that all the animations that in Experiment 1 were associated with clear physical bounce impressions were still associated with clear physical bounce impressions in Experiment 3. A similar reasoning can be applied to animated motion and "other" impressions, and to Fig 8.

R8: Lines 471ff: The time it takes a tennis ball to deform and rebound (5 ms) may not be a convincing argument that the delay times of 30 ms and above are impossible. Take a large ball that is not inflated to its maximum. I bet that rebound times around 30 ms are possible. This aside, the visual system integrates sudden onset movement over even larger time windows (see e. g. Runeson, S. 1974, Constant velocity—not perceived as such. Psychological Research, 37(1), 3-23). Would the authors argue that this is what happens as well in the case of bouncing events?

 AU8: We thank the Reviewer for giving us the opportunity to discuss this points in greater detail. We have explicitly acknowledged and discussed the possibility that delays of 30 ms or longer can be naturally occurring in certain circumstances (see lines 626-631 of the manuscript with track changes). In this regard, it is important to note that the stimuli used in our experiments did not exhibit any noticeable deformation upon impact with the bouncing surface, which makes them more similar to rigid steel balls than to underinflated rubber balls. Regarding the relationship between the results from Experiment 2 and the findings of Runeson’s (1974) study, we believe that a reliable comparison between the sudden motion onset of an initially stationary object and the direction change of a bouncing object is not feasible due to the stark differences between the two situations from a perceptual standpoint. The key disparity is that while the motion onset of an initially stationary object is unpredictable, the direction reversal of a bouncing object is fully predictable based on the position of the bouncing surface.

R9: The sentence starting on line 871 “We speculate …” is awful and flawed. “Cue of the physical plausibility” makes no sense.

 AU 9: Thank you for having pointed this out. The flawed sentence have been rephrased as follows: “We speculate that, for animations depicting only one bouncing cycle, the visual information regarding parameter C might not have been presented redundantly enough to function as a reliable indicator of its physical plausibility” (see lines 1243-1245 of the manuscript with track changes).

R10: Line 887: The sentence starting with “From a physical viewpoint bouncing is a unitary event …” is misleading. A falling object is a lot simpler to describe than a bouncing object. I guess what is meant here is that a truncated event, i. e. one bounce only, may look odd although it simpler than the complete event.

 AU10: What we meant to say is that the physical laws underlying bouncing events characterized by one bouncing cycle are the same as those underlying bouncing events characterized by three bouncing cycles. We have rephrased this misleading sentence as follows: “From a physical viewpoint, all the relevant physical laws are independent of the number of bouncing cycles” (see lines 1261-1262 of the manuscript with track changes).

R11: Line 909: The statement that acceleration changes as a function of distance to the event is incorrect. Just because the visual angle (and retinal size) of an object changes with distance, the acceleration does not. A Martian g would be very strange for a terrestrial environment, no matter how far away the observer is from the falling object.

 AU11: We have rephrased the whole passage so to increase the clarity and the transparency of its meaning (see lines 1284-1291 of the manuscript with track changes): “It is worth noting that the physical acceleration of a falling object can produce different retinal acceleration values depending on the object's distance from the observer. As per the inverse relationship between retinal acceleration and distance [72], a small object located nearby and falling with a physically unrealistic low acceleration value could produce the same retinal acceleration as a large object located far away falling with a physically plausible acceleration of g. Given that our stimuli lacked absolute cues for size and distance, it is possible that the observers could have perceived accelerations of g/4 and g/16 as instances of large distant objects falling with a physically accurate acceleration of g, instead of a small object close to the observer's viewpoint falling with an implausible physical acceleration.”

We would like to express our appreciation for the Reviewer's painstaking work and the significant insights he provided, which have undoubtedly contributed to improving the manuscript. Thank you once again for your time and valuable contribution to our work.

RESPONSES TO THE COMMENTS FROM REVIEWER #2

R1: This study reports 4 experiments investigating which visual parameters (kinematics, presence or not of a bouncing surface) produce physical or animacy (or neither) interpretations of a bouncing motion. Although I found most of the experimental manipulations relevant and I may agree with the theoretical framing used for interpreting the findings, I think as it stands, the statistical analyses do not do the readers any favors or do the data justice in order to help draw out the essence of the data. That is why the bulk of my report will focus on these analyses, and I recommend a major revision.

 AU1: We would like to extend our sincere appreciation to the Reviewer for the time taken to re-analyse some of our data, and for the valuable and detailed feedback regarding how to enhance the statistical analyses and presentation of the results. We have taken her/his recommendations into account, and the data from the four experiments have been re-analysed using a repeated measures ANOVA approach, as per the Reviewer's suggestion. The results sections have been revised accordingly (for detailed explanations see our responses below).

R2: First, I must confess that I had great difficulty in following the exposure of results starting with the Box and Whisker plots used in Figure 5 (Expt1) and 6 (Expt2). By the way, the authors abandoned such Figures in the subsequent experiments. Why use these nonparametric plots to illustrate results from parametric Generalized Linear Mixed Models (GLMM, here, with a logit link function for the logistic regressions)? GLMM typically illustrate Fixed and random effects in a very different way than in the manuscript. I give some illustrations in Appendices 1-3 (after all my comments), for Experiment 2 data that I downloaded from the link provided in the manuscript. When predictors are treated as “factors” then, the figures look like ANOVA figures (see Appendix 1 and 3). Moreover, in this case the random effects plotted by participant should be added to the figures as separate lines in order to better visualize the factors effects at the individual, which cannot be visualized in the kind of plots actually used by the authors. In contrast, if some experimental factors are treated as continuous variables (covariates), the predictor effect may be visualized using logistic curves, or with linear functions after logit transform of the dependent variable (see Appendix 2). Examples of these figures can be found in: https://gamlj.github.io/glmmixed_example1.html#The_research_design

 AU2: We would like to express our gratitude to the Reviewer for the valuable suggestions on how to effectively present the results obtained from GLMM models. We will undoubtedly apply them to our future studies. As for the present manuscript, based on the Reviewer's feedback, we have decided to abandon the GLMM approach and adopt the repeated measures ANOVA approach with Huynh-Feldt correction. Consequently, we have taken the following steps: 1) relocated Figures 2, 3, and 4 to OSF, where they will serve as supplementary figures; 2) removed Figures 5 and 6 (the box and whisker plots); 3) presented the results from Experiment 1 and 2 using line plots (new Figures 2, 3, and 4) similar to those in Appendix 5 of the Reviewer’s report. We hope that these changes will improve the visualization of the results and enhance the consistency between the statistical approach and the related graphical representations.

 R3: Moreover, the reason for using GLMM (sometimes also called MLM, multi level models) instead to ANOVA is unclear/unjustified. The authors just refer Jaeger (2008, doi: 10.1016/j.jml.2007.11.007) paper who promoted using Logit Mixed Models instead of ANOVAs over proportions of categorical outcomes. Among others, one argument was “even ANOVAs over arcsine-square-root transformed proportions of categorical outcomes (see below) can lead to spurious null results and spurious significances. These spurious results go beyond the normal chance of Type I and Type II errors. “ (Jaeger, 2008, p.435). However, recent papers clearly challenge Jaeger (2008) conclusions with simulation studies that plead for the use of repeated-measures ANOVA with Huynh-Feldt-correction (especially when the sphericity assumption is violated), rather than multi-level models, when the sample size is rather small and the number of measurement occasions is large (Haverkamp & Beauducel, 2017, DOI: 10.3389/fpsyg.2017.01841). Likewise, Luepsen (2021, DOI: 10.1080/03610918.2020.1869983) simulation study showed that for binary variables, “Depending on the design and on the model to be checked, in most cases the parametric F-test with adjustment, the multivariate test or Koch’s method are advised”, as compared to GLMM and GEE (Generalized Estimating Equations).

 AU3: We are grateful to the Reviewer for these fundamental references, which challenged our previous opinion about the suitability of GLMM to analyse dichotomous data. We have cited the works mentioned by the Reviewer in the revised manuscript (see lines 426-433 of the manuscript with track changes), re-analysed the data according to the repeated measures ANOVA approach with Huynh-Feldt-correction, and rewrote the results sections accordingly. This is a brief summary of our revised approach to the analysis of response proportions:

 1) Experiment 1: We first conducted three separate four-way ANOVAs on response proportions (one for each response category), with factors Number of bounces, Motion pattern, a, and C. The results of these analyses are reported in Supplementary Tables S1, S2, and S3 on OSF (https://osf.io/xz8t3/?view_only=741328b1d38b4f8d9d72bbfbe745ba85). Following the results of these analyses, we then conducted separate ANOVAs for one and three bouncing cycles, separately for each response category. The results for the physical bounce and animated motion responses are reported in Table 1 and 2, whereas the results for the other responses are reported in Supplementary Table S4. 

 2) Experiment 2: A detailed summary of the analysis plan is presented on lines 682-699 of the manuscript with track changes. In brief, we first conducted three-way ANOVAs on each response category, with factors Number of bounces, C, and Delay. The results for physical bounce and animated motion responses are reported in Table 3, whereas the results for other responses are in Supplementary Table S7. Then, we conducted one-way ANOVAs on each response category, with Delay as factor, separately for each animation type. The results for physical bounce and animated motion responses are reported in Table 4, whereas the results for other responses are in Supplementary Table S7. Post-hoc comparisons between delay levels (with Bonferroni correction) are reported in Supplementary Tables S5, S6, and S7. Lastly, we compared the probability of physical bounce and animated motion responses at each delay level for the four animation types, using paired-sample t-tests with Bonferroni correction. We present the results of this analysis in Table 5. 

 3) Experiment 3: We conducted three separate two-way mixed ANOVAs, one for each response category, on response proportions, with Experiment (Experiment 1 or Experiment 3) as a between-subject factor and the number of bouncing cycles (one or three) as a within-subject factor. The ANOVA results are presented in Table 6.

 4) Experiment 4: We first conducted three separate two-way mixed ANOVAs (one for each response category) with Surface (opaque surface or amodal surface) as a between-subject factor and the number of bouncing cycles (one or three) as a within-subject factor. The results, presented in Supplementary Table S8 on OSF, indicate no statistically significant main or interaction effects for the Surface factor. Then, we performed three two-way mixed ANOVAs, one for each response category (physical bounce/external physical cause, animated motion/internal psychological cause, other), with Experiment (Experiments 1/2 or Experiment 4) as a between-subject factor and the number of bouncing cycles (one or three) as a within-subject factor (no distinction between visible and amodal surface). The ANOVA results are presented in Table 7.

R4: What’s more, the way authors use GLMM is unconventional. The standard approach is first to test the intercept only model (Null model), to check if there is substantial evidence of clustering in the data to justify a multilevel approach. In fact, this does not seem to be the case in Experiment 2 for the “physical” response dependent variable (see the Tables below) as suggested by the ICC (intraclass correlation) values that are smaller than 0.05 which is a usual cut-off value1 above which to go on with GLMM approach (when ICC > 0.05). ICC quantifies the proportion of variance explained by a grouping/random factor in multilevel data, and corresponds to the R² conditional value in the null model.

 AU4. We are grateful to the Reviewer for these suggestions, which we will apply in future studies with GLMM.

R5: Likewise, instead of listing all the comparisons and associated odds ratio (OR), these OR could be visualized in a figure (see:https://strengejacke.github.io/sjPlot/articles/plot_model_estimates.html) 

with the interpretation summarized in a few sentences without the p values (these could be provided in Supplementary Material). The idea, here, is to facilitate the reading of results that are presented in a rather choppy way for the moment. Jtools is perfect for summarizing and visualizing regression models (https://jtools.jacob-long.com/articles/summ.html)

 AU5. To enhance readability and clarity in the revised manuscript, we aimed to limit the number of F-values, p-values, and other statistical measures included in the main text. Rather, we provided a comprehensive description of the primary results with a focus on the conceptual understanding of the findings. Tables included in the manuscript and supplementary tables on OSF were utilized to report detailed statistical analyses. 

R6: Furthermore, I don’t understand why interaction terms were not introduction in the GLMM models of the authors. Actually, one should start with a null model to test if the GLMM approach is justified (by looking at ICC of the null model) and if so, then increase the complexity of the model up to the point of including interactions. Then redundant predictors need to be identified (looking at correlations) and deleted from the model, until a fair tradeoff between explained variance and complexity of the model is met. This is not at all the way the authors use GLMM. For examples of Logistic Mixed Effects Model with Interaction Term and Logistic Mixed Effects Model with Three-Way Interaction (with lme4, glmer): 

• https://strengejacke.github.io/ggeffects/articles/practical_logisticmixedmodel.html

• Hox (2010) Multilevel Analysis Techniques and Applications Second Edition. Routledge 

• See also Mike Crowson tutorials: https://www.youtube.com/watch?v=8r9bUKUVecc

Instead the authors chose to perform multiple GLMM sub-analyses, that are equivalent to performing t tests (actually the authors used z scores and odds ratio) on multiple paired comparisons to decompose an ANOVA interaction, without showing first that the interaction is significant!

 AU6: We acknowledge and appreciate the Reviewer's feedback on the unconventional use of GLMM in our original manuscript. While our intention was to simplify the analyses and focus solely on the primary results, we recognize that this approach may have led to confusion and misinterpretation of the results. Therefore, we have made significant revisions to the manuscript and have opted to report ANOVA results in a conventional manner. Specifically, we conducted ANOVAs with all experimental factors and followed up with sub-analyses on specific factors when their interactions were statistically significant. We believe that this revised methodology will increase the clarity of the results.

R7: Also, the pairwise analysis is not homogenous across the experiments. For example, the authors performed paired comparisons between adjacent delay levels (without controlling for multiple testing!) in Experiment 2, but not for the a and C levels in Experiment 1.

 AU7: We appreciate the Reviewer's concerns about the heterogeneity of pairwise analyses across experiments. In Experiment 2, we conducted specific paired comparisons within the same response category for adjacent delay levels and between response categories for identical delay levels. To adhere to the standard approach to post-hoc comparisons, in the revised manuscript we have now performed paired comparisons within the same response category for all delay levels (not just adjacent levels), as shown in Supplementary Tables S5, S6, and S7. Bonferroni correction has been applied in each case. We consider these comparisons important for our primary hypotheses and believe that the relatively small number of comparisons makes the results easily interpretable for readers. Conversely, in Experiment 1, we did not deem pairwise comparisons between C levels necessary. Not only would such comparisons not significantly alter the main conclusions, but the vast number of potential comparisons would make interpretation difficult for readers. Similarly, we believe that pairwise comparisons in Experiments 3 and 4 would not offer significant insights into the main results. Overall, we have chosen to focus on the main effects and interactions of our experimental factors while providing the necessary post-hoc comparisons only when this was important to test our primary hypotheses.

R8: In addition, stimuli could have been treated as random factors (see Judd et al 2012 DOI: 10.1037/a0028347) and interaction between the random effects studied, from the start in Expt1 and 2. Thereby level of “C” or “delay” factors would be considered just samples of possible stimuli out of a larger variation in levels of stimuli. Unfortunately, authors waited Expt3 and 4 before touching on the possibilities offered by this approach but without reporting/discussing the variances nor the ICC values nor the correlations between random effects the added value of these analyses (as compared to not including stimuli as random effects) is simply lost. Finally, why not using random intercept and slope models, that would possibly increase the explanatory power of GLMM, as suggested by Appendix 4 for an example for their Expt2. Likelihood Ratio Test statistic (LRT) is also useful to test if larger models provide significant improvement over reduced models (see Roback & Legler, 2021, https://bookdown.org/roback/bookdown-BeyondMLR/ ), and also to analyze the variance components of the model. Regarding model comparisons, see : Raudenbush, S. W., & Bryk, A. S. (2002). Hierarchical linear models: Applications and data analysis methods (2nd ed.). Newbury Park: Sage.

 AU8: We thank the Reviewer for the insightful suggestions. We did not implement these suggestions in this revised manuscript only because we decided to switch to a repeated measures ANOVA approach, as per the Reviewer’s indications.

R9: In sum, given all the above issues, instead of using a GLMM approach I would rather suggest that the authors use Huynh-Feldt corrected ANOVAs, and interpret the parameters of the associated psychometric functions (instead of testing multiple pairwise comparisons to each stimulus level of the psychometric function), starting with an omnibus ANOVA on response proportion using the different response categories are levels of a factor, as in the Appendix 5 example for their Expt2. Then, onmnibus ANOVAs may be run on each separate response type (provided that response category interacted with the other within-subject factors in the initial ANOVA), followed by sub-ANOVAs for decomposing complex interactions. Additionally, the authors could either make an ANOVA on the psychometric lapse rate, guess rate, PSE, and JND parameters (see Kingdom & Prins, 2010, Psychophysics: A practical introduction book) data for each individual, or describe the above mentioned ANOVA interactions on the basis of the psychometric parameters computed from the mean response proportions (see examples below) rather than use pairwise comparisons. Note that PSE (transition point between response categories) and JND parameters are the main parameters of interest. For example, according to the Figure below, the interaction between C and Delay (F(5, 145)= 11.13, p < 0.0001, H-F epsilon = 0.83) on the Proportion of “physical responses” in Experiment 2, reflects similar transition point (about a 68 ms delay) from physical to animated&other responses for both C levels, but a greater change in response for C = 0.75 compared to C = 1.25 (see JND, lapse and guess parameters). Note that the authors did not focus on these transition values (PSE), neither for delay in Expt 2, nor for C in Expt1, although they summarize the psychophysical effect of the experimental manipulations.

 AU9. We thank the Reviewer for offering us this alternative perspective on the analysis and the interpretation of the results of Experiments 1 and 2. While we agree with the general idea that the transition between physical bounce and animated motion responses and the differences in observers' sensitivity to the manipulation of C and Delay are important, we respectfully disagree that "PSE and JND parameters are the main parameters of interest."

 First, PSEs and JNDs would provide ambiguous (i.e., not easily interpretable) information in several conditions of Experiment 1, because for physical bounce responses for one bouncing cycle the data distribution is inconsistent with a psychophysical function (see Figure 2). Second, in Experiment 2, PSEs and JNDs would not provide all the essential empirical results, such as the slight increase in the proportion of physical bounce or animated motion responses with a 30 ms delay, the steep decrease in the probability of physical bounce or animated motion responses for delays longer than 30 ms, and the lack of a significant difference in the proportion of physical bounce and animated motion responses with three bouncing cycles and C = 0.75 for delays up to 90 ms. Therefore, we believe that solely comparing PSEs and JNDs across different animation types would result in missing some critical theoretical results that require a discussion of pairwise comparisons across different delay levels for the same response category and across different response categories for the same delay.

 However, we agree with the Reviewer that the transition between physical bounce and animated motion responses, even when it is not present, as in the case of one bouncing cycle in Experiment 1, is important to discuss. We also agree that the sensitivity of observers to changes in parameters C and delay should be emphasized, particularly for different combinations of bouncing cycles and C. We have commented on these points in the revised manuscript (i.e., transition between physical bounce and animated motion responses: see lines 535-537, 546-547; varying magnitudes of the effects of C and delay on each response category: see lines 526-552 and lines 732-772, respectively). We believe that these descriptions provide the reader with all the relevant theoretical information, despite the absence of quantitative information about PSEs and JNDs. 

 We also want to make it clear that we did not include response category as a factor in the ANOVAs because physical bounce and animated motion are two concepts that are negatively related. When physical bounces are perceived as clear, it is less likely for animated motion to be perceived, and vice versa. Additionally, participants had to choose from three response alternatives, so a high probability of one response category corresponded to a low probability of the other two categories. Therefore, we believe that the interaction between the response category and other manipulated factors in each experiment could be safely assumed a priori, for both theoretical and methodological reasons. There was no need to test it with inferential statistics. The graphical representations illustrate that the behavior of physical bounce and animated motion impressions was approximately specular.

R10: Likewise, the multiple pairwise analyses of Experiment 1 also boil down to the logistic psychometric parameters whether using CDF or PDF. For example, the interaction between C and Motion pattern on the Proportion of “physical responses” (F(8, 232)= 8.14, p < 0.0001, H-F epsilon = 0.832) illustrated below for the 3 bouncing cycles condition simply shows that the transition point from physical to animated&other responses is much earlier for the uniform acceleration motion pattern (about C= 0.63) as compared to the uniform acceleration motion pattern (about C= 0.81).

 AU10: We believe that this relatively small difference between the transition point for uniform acceleration and uniform velocity does not require explicit discussion within the complex picture of the results. Instead, it can be viewed as a consequence of the more general effect of the motion pattern factor. Specifically, physical bounce responses were more likely in the case of uniform acceleration than in the case of uniform velocity, regardless of whether there was one or three bouncing cycles (although the effect was clearly stronger for one bounce than for three bounces). Uniform acceleration facilitated the perception of a physical bounce compared to uniform velocity, and it is natural that the effect was most evident for the values of C that were more likely to give rise to a physical bounce response (which explains the Motion pattern × C interaction). Because of this, the curve for physical bounce responses fell below the .5 threshold earlier for uniform velocity than for uniform acceleration. Therefore, we believe that this small difference is not a significant finding in itself but rather a result of the broader effect of the motion pattern factor.

R11: In Experiment 1, from the results for the “varying a” and the “varying C” animations (see Fig. 4), authors conclude that "None of these animations gave rise to clear physical bounce or animated motion impressions." In order to make the transition between Expt1 and 2 clearer, I suggest to move the exposure of these experimental manipulations in a Supplementary material section.

 AU11: To provide a comprehensive view of our findings, we would like to emphasize that the quoted passage pertains solely to the results of the "varying a" animations. In contrast, for the "varying C" animations, we reported that "Only two of these animations gave rise to clear animated motion impressions; however, most of these animations had a probability close to .50" (see lines 609-610 of the manuscript with track changes). Since these animations were part of the primary experimental design, and the results for the "varying C" animations are quite clear, we believe it is necessary to include a brief discussion of these findings in the main text. However, the old Figure 3, which represented the results for these animations, has now been moved to the supplementary materials (Fig S3 on OSF).

 R12: Likewise, I wonder if the analysis of “other” responses should be included in the main text. First, these responses were not analyzed in Experiment 2 (due to their low frequency). Second, this response category is not central for the discussion of the study. I suggest to move the data analysis for “other” responses in the Supplementary Material. As a consequence, it could therefore be removed from omnibus ANOVAs in the main text, if one starts with a “response category” factor with two levels (physical, animated) such as in the Appendix 6 example.

 AU12: We would like to thank the Reviewer for this comment. We have incorporated the suggestion in the revised manuscript by moving the analyses of other responses for Experiment 1 to the supplementary materials (Supplementary Tables S3 and S4). To ensure completeness and consistency with Experiment 1, we have also analyzed other responses in Experiment 2. The average probability of other responses is now presented in Fig 4, and the results of statistical analyses, including the omnibus ANOVA, specific ANOVAs for each animation type, and post-hoc comparisons, are reported in Supplementary Table S7.

AU 13: Lastly, we would like to thank the Reviewer for the clarity of the examples provided by figures and tables in the appendixes of his/her report, particularly on the representation of random effects. In response, we have made several changes in the revised manuscript. To enhance the visual clarity of the results, we have represented the data from Experiments 1 and 2 using line graphs, displaying the confidence intervals around each curve. We have also adopted tables similar to those reported by the Reviewer in Appendixes 5 and 6 to represent the results of both experiments. 

We would like to express our appreciation for the Reviewer's painstaking work and the significant insights provided, which have undoubtedly contributed to improving the manuscript. Thank you once again for your time and valuable contribution to our work.

---

## [Decision Letter · Decision Letter 1]

24 Apr 2023

The psychophysics of bouncing: perceptual constraints, physical constraints, animacy, and phenomenal causality

PONE-D-22-33138R1

Dear Dr. Parovel,

We’re pleased to inform you that your manuscript has been judged scientifically suitable for publication and will be formally accepted for publication once it meets all outstanding technical requirements.

Kind regards,

Robin Baurès, Ph.D.

Academic Editor

PLOS ONE

Additional Editor Comments (optional):

Dear Dr. Parovel,

As you will read, both authors found that you have made a very nice job when revising your paper and taking their comments into account. I am really pleased to suggest its acceptance. There remains however one last suggestion made by reviewer 1 (Heiko Hecht), but I don't think it is worth another round of review. I would rather suggest you see how to deal with it in the final (editing) steps.

Congratulation on this nice work!

Best,

Robin

Reviewers' comments:

Reviewer's Responses to Questions

**Comments to the Author**

1. If the authors have adequately addressed your comments raised in a previous round of review and you feel that this manuscript is now acceptable for publication, you may indicate that here to bypass the “Comments to the Author” section, enter your conflict of interest statement in the “Confidential to Editor” section, and submit your "Accept" recommendation.

Reviewer #1: All comments have been addressed

Reviewer #2: All comments have been addressed

2. Is the manuscript technically sound, and do the data support the conclusions?

Reviewer #1: Yes

Reviewer #2: Yes

3. Has the statistical analysis been performed appropriately and rigorously? 

Reviewer #1: Yes

Reviewer #2: Yes

4. Have the authors made all data underlying the findings in their manuscript fully available?

Reviewer #1: Yes

Reviewer #2: Yes

5. Is the manuscript presented in an intelligible fashion and written in standard English?

Reviewer #1: Yes

Reviewer #2: Yes

6. Review Comments to the Author

Reviewer #1: Review of revised manuscript “The psychophysics of bouncing: perceptual constraints, physical constraints, animacy, and phenomenal causality”

The authors have done a commendable job replying to the reviewers’ comments and addressing the issues in their revision. I can generally follow the arguments and get the impression that the manuscript has greatly improved. All the points I raised in my initial review have been adequately dealt with. The paper will make a valuable addition to our knowledge about the intuitive physics of bouncing balls. I merely have a few minor remaining points I would like to raise:

Line 11 (numbers refer to the pdf) in the Abstract: “of” is missing -> coefficient of restitution

Line 1035: The fact that observers underestimate acceleration does not necessarily imply that they generally prefer accelerations smaller than g. It could just as well mean that they cannot handle fast retinal velocities and prefer retinal velocities that are moderate, not too fast and not too slow. When a far-away sky-scraper falls, it looks ridiculously slow and does not seem to accelerate properly. It is my guess that observers, when they watch moving objects, attach labels to them that are based first and foremost on velocity impressions, not on acceleration impressions. And these are calibrated relative to our normal action space and not to the far away vista space. Thus, the impression of “unnaturally fast” or “unnaturally slow may reflect velocity preferences rather than acceleration preferences, as the authors suggest. Here it may be appropriate to flag their preference as such. The results seem to be compatible with both views.

As an aside, to support may interpretation, we asked subjects in a paper-and-pencil task which of several broom-sticks would fall faster when toppled over and would thus be harder to balance, sticks with a weight mounted at the top or sticks with the weight at the bottom. They performed very poorly, suggesting that even for objects at close range, our intuitions about acceleration are quite erroneous (see Hecht, H. (2015). Beyond illusions: On the limitations of perceiving relational properties. https://openscience.ub.uni-mainz.de/bitstream/20.500.12030/663/1/55023.pdf).

Signed review

Heiko Hecht

Reviewer #2: I very much appreciate the manuscript as it stands in its revised form, and I highly recommend its publication.

7. PLOS authors have the option to publish the peer review history of their article (what does this mean?). If published, this will include your full peer review and any attached files.

Reviewer #1: No

Reviewer #2: **Yes: **Michel-Ange Amorim

---

## [Editor Report · Acceptance letter]

8 May 2023

PONE-D-22-33138R1 

The psychophysics of bouncing: perceptual constraints, physical constraints, animacy, and phenomenal causality 

Dear Dr. Parovel:

I'm pleased to inform you that your manuscript has been deemed suitable for publication in PLOS ONE. Congratulations! Your manuscript is now with our production department. 

Kind regards, 

on behalf of

Dr. Robin Baurès 

Academic Editor

PLOS ONE